# Finding landmarks - An investigation of viewing behavior during spatial navigation in VR using a graph-theoretical analysis approach

**Jasmin L. Walter**[1☉]*, **Lucas Essmann**[1☉], **Sabine U. König**[1], **Peter König**[1,2]

**1** Institute of Cognitive Science, University of Osnabrück, Osnabrück, Germany, **2** Department of Neurophysiology and Pathophysiology, University Medical Center Hamburg-Eppendorf, Hamburg, Germany

☉ These authors contributed equally to this work.
* jawalter@uni-osnabrueck.de

**Data Availability Statement:** All data files the described pre-processing and analysis are based on, are available at the Center for Open Science https://osf.io/aurjk/, DOI 10.17605/OSF.IO/AURJK.

## Abstract

Vision provides the most important sensory information for spatial navigation. Recent technical advances allow new options to conduct more naturalistic experiments in virtual reality (VR) while additionally gathering data of the viewing behavior with eye tracking investigations. Here, we propose a method that allows one to quantify characteristics of visual behavior by using graph-theoretical measures to abstract eye tracking data recorded in a 3D virtual urban environment. The analysis is based on eye tracking data of 20 participants, who freely explored the virtual city Seahaven for 90 minutes with an immersive VR headset with an inbuild eye tracker. To extract what participants looked at, we defined "gaze" events, from which we created gaze graphs. On these, we applied graph-theoretical measures to reveal the underlying structure of visual attention. Applying graph partitioning, we found that our virtual environment could be treated as one coherent city. To investigate the importance of houses in the city, we applied the node degree centrality measure. Our results revealed that 10 houses had a node degree that exceeded consistently two-sigma distance from the mean node degree of all other houses. The importance of these houses was supported by the hierarchy index, which showed a clear hierarchical structure of the gaze graphs. As these high node degree houses fulfilled several characteristics of landmarks, we named them "gaze-graph-defined landmarks". Applying the rich club coefficient, we found that these gaze-graph-defined landmarks were preferentially connected to each other and that participants spend the majority of their experiment time in areas where at least two of those houses were visible. Our findings do not only provide new experimental evidence for the development of spatial knowledge, but also establish a new methodology to identify and assess the function of landmarks in spatial navigation based on eye tracking data.

All pre-processing, visualization and analysis scripts applied in this publication are available on our Github repository including an extensive documentation to allow full reproducibility: https://github.com/JasminLWalter/FindingLandmarks_a_publication_repository. In addition, we have archived the repository using Zenodo and assigned a persistent identifier to it: DOI: 10.5281/zenodo.6515988.

**Funding:** The project was financed by the Research Training Group Computational Cognition, which is funded by the Deutsche Forschungsgemeinschaft (DFG, German Research Foundation) – GRK2340 and the Open Access Publishing Fund of Osnabrück University. One of the first authors, JLW, is supported by a position in the Research Training Group. The funders had no role in the study design, data collection and analysis, decision to publish, or preparation of the manuscript.

**Competing interests:** The authors have declared that no competing interests exist.

## Author summary

The ability to navigate and orient ourselves in an unknown environment is important in everyday life. To better understand how we are able to learn about a new environment, it is important to study our behavior during the process of spatial navigation. New technical advances allow us to conduct studies in naturalistic virtual environments with participants wearing immersive VR-headsets. In addition, we can use eye trackers to observe the participant's eye movements. This is interesting, because observing eye movements allows us to observe visual attention and, therefore, important cognitive processes. However, it can be difficult to analyze eye tracking data that was measured in a VR environment, as there is no established algorithm yet. Therefore, we propose a new method to analyze such eye tracking data. In addition, our method allows us to transform the eye tracking data into graphs which we can use to find new patterns in behavior that were not accessible before. Using this methodology, we found that participants who spend 90 min exploring a new virtual town, viewed some houses preferentially and we call them gaze-graph-defined landmarks. Our further analysis reveals also new characteristics of those houses that were not yet associated with landmarks.

## Introduction

Having a sense of orientation and being able to navigate in the world that surrounds us is essential in everyday life. Specifically, the awareness of one's own position in space combined with the ability to remember key locations to plan mental routes between them [1] is crucial. This enables efficient navigation to a location by using globally accessible knowledge of a new environment or previously acquired knowledge of a known environment. Overall, the ability to remember and use important locations and their relations is essential for spatial navigation.

The impact of remembered objects in the environment, called landmarks, on spatial navigation abilities is widely investigated. Even though it lacks a clear definition, the term landmark is most commonly used for visually salient objects in the environment [2]. Landmarks can be distal objects that can be seen from far away, termed global landmarks, or more proximal objects that serve as local landmarks [3]. Single landmarks can be used as navigational beacons, e.g., a church tower [2,4,5] and can also provide visual cues for direction and orientation information [2,6,7]. Getting to know an environment, environmental objects develop action-relevant associations, like turn left at the supermarket, for wayfinding to a goal [2,8,9]. Furthermore, multiple landmarks and their intrinsic geometry can be used as an allocentric reference frame for object location information [2,4,10,11]. Thus, visual landmarks and their multiple function in everyday navigation, have been cornerstones for spatial navigation.

In spite of their ubiquitous use, investigating the concept of landmarks does not come without problems. Often, studies investigate landmark based navigational learning by introducing single cues to serve as landmarks in an otherwise undifferentiated environment. For example, the Morris water-maze is a widely used task to study the physiological mechanisms of spatial learning in rodents [12–14]. Similarly, adaptations of the Morris water-maze in virtual environments [15–17] and other maze tasks [18,19] have been used to investigate spatial navigation in humans. However, in recent years experimental designs of spatial navigation research have shifted to make use of more naturalistic settings [20–27]. The trend towards more naturalistic environments can also be observed in virtual environments, where technical advances allow for more complex and visually rich environments [28,29], as well as increased ecological validity including freedom of movement using head mounted virtual reality headsets [30–32].

This development towards more complex environments creates the need to clearly define the concept of landmarks.

In addition, performing eye tracking during spatial navigation also allows capturing visual behavior in real-time, and investigating the most important sense for spatial navigation in humans [33]. While there are studies investigating visual behavior in real world indoor [22,23] and outdoor [24–27] environments, they are often challenged by limited experimental control due to variance in environmental conditions [24–26] and technical drawbacks of mobile eye tracking like manual frame-based detection of regions of interests [22,26,27] and limited calibration options [26]. In contrast, technological advances in VR technology often allow the direct integration of eye trackers into head-mounted VR headsets [34], therefore facilitating eye tracking recording in complex and more ecologically valid environments while maintaining full experimental control. However, although eye tracking in VR facilitates technical aspects related to the calibration and localization of regions of interests [34], it still creates new challenges for the analysis of eye tracking data. In classical visual exploration of static 2D images, fixations and saccades dominate. In a complex 3D environment vestibulo-ocular reflexes and pursuit movements additionally occur on a regular basis. However, there is no established algorithm to differentiate this expanded set of eye movements in eye tracking data collected in a 3D environment. Thus, to identify characteristics of visual behavior during free exploration of a virtual village and how they relate to spatial navigation, a new method is needed.

In this study, we propose a procedure to analyze eye tracking data and a data driven method to objectively define and quantify visual behavior with respect to spatial information. We use a graph-theoretical analysis approach to access global navigation characteristics across participants and investigate the occurrence, connectivity, and navigation function of a subset of houses, consistently outstanding in their graph-theoretical properties. The approach is applied to eye tracking data recorded during free exploration of a virtual town that was originally designed to investigate spatial learning within an immersive virtual environment in comparison with learning of a city map of the same environment [35,36]. Overall, our findings establish a new methodology to identify and assess the function of outstanding houses in spatial navigation based on eye tracking data.

## Results

We collected eye tracking data of 22 participants (11 females, age: M = 22.86y, SD = 2.59y) during 90 min of free exploration in the virtual city Seahaven. All participants gave their written informed consent to participate. The study was approved by the Ethics Committee of the Osnabrück University following the Institutional and National Research Committees' ethical standards. The participants wore a VR headset with an inbuild eye tracker. They moved using a controller and physically rotated their body on a swivel chair to turn in the virtual world (Fig 1A). The virtual city was built on an island and comprised 213 houses (Fig 1B). Colliders, i.e., transparent closely fitted box-like structures, surrounded all houses, trees, and roads. They are invisible to the participants and serve to facilitate computations in the virtual environment, e.g., detecting collisions and determining viewing direction. Specifically, colliders are components in Unity that can be fit to an object to identify the object's physical boundary. Colliders therefore provide the basis for the system calculating properties of the real world like an object being solid and therefore "colliding" with other solid objects. Furthermore, we calculated the viewing direction based on the position of the participant in the VR, the rotation of the headset, and information from the eye tracker. After data collection, we cast a virtual ray in the viewing direction until it hit a collider, thus the boundary of an object, indicating that the

**a.**

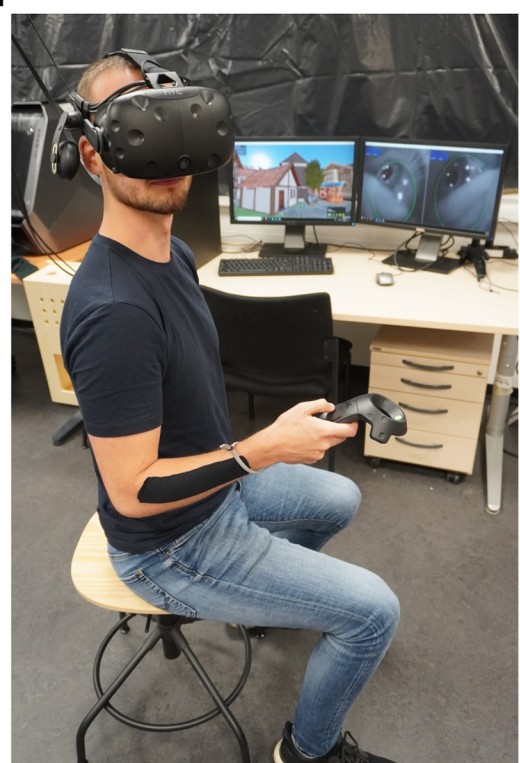

**b.**

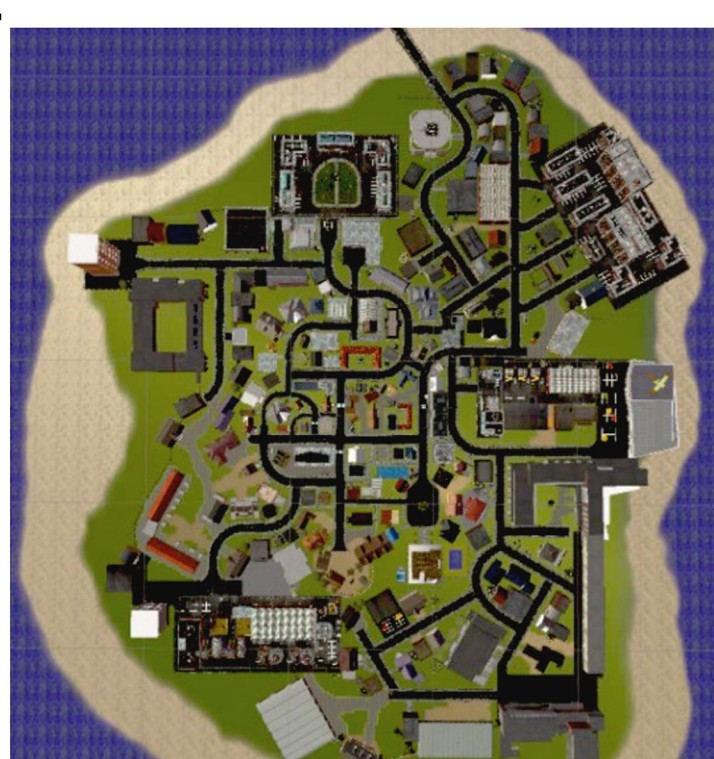

**Fig 1. Experimental Setup and Seahaven. (a)** The participant sits on a fully rotatable swivel chair wearing headset and controller. The experimenter's screens in the background allow monitoring the participant's visual field of Seahaven (left) and the pupil labs camera images (right). **(b)** The island of Seahaven in aerial view.

participant viewed the respective object [34]. This process was completed 30 times per second. Aggregated over all participants, this led to about 3.500.000 collider hit points that form the basis of our data analysis.

## Defining gazes in VR

Eye tracking in an VR environment provides several challenges for data quality. Compared to conventional desktop settings, VR environments allow increased freedom of movement. Under these conditions, participants perform fixations, saccades, vestibulo-ocular reflexes, and smooth pursuit movements. No general algorithm to classify these types of eye movements has been established yet. Thus, the development of appropriate processing algorithms for the eye tracking data is crucial.

As a first step, we use the collider hit points resulting from casting a ray in viewing direction until it hits an object's collider to identify the object the participants focused their gaze on. Each collider corresponds to an entire object; thus, each hit point identifies an entire object. For our analysis, houses form the regions of interests (ROIs). The "no house" category (NH) summarizes all other collider hits except houses, e.g., grass, roads, trees, and the water. Samples that do not hit a collider, thus hitting the sky or the sun, are identified as "sky" category. We then combine directly consecutive hit points on the same collider to identify clusters. Please note that the hit point clusters do not contain information on where in the participant's visual field the viewed object occurred. This pre-processed data of combined consecutive hit points on the same collider serves as the basis for all following data processing.

As a second step, we address the problem of missing data in individual participants. We label all data samples that were recorded with a probability of correct pupil detection of less than 50% as "missing data samples". Subsequently, we exclude two participants who had more than 30% of their eye tracking data classified as missing data. All further data analysis is conducted with the data of the remaining 20 participants.

In the following we will need to differentiate between periods where participants could perceive the visual stimuli or not. Given that vestibulo-ocular reflexes and smooth pursuit movements stabilize the retinal image in dynamic situations and allow perception in that period similar to fixations [37], here and in all following analyses we subsume these under the general term of fixations. In contrast, participants are blind to visual input during saccadic suppression [37,38]. While classical fixation detection algorithms often differentiate eye movements based on velocity, these eye movements also display a temporal disparity. Specifically, saccades usually range from a duration of 10 to 100 ms, while fixations typically occur from 150 to 600 ms with a mode around 250 ms [37]. Therefore, we conjecture that with an appropriate temporal threshold, it is possible to identify data clusters containing fixations. Based on our 30 Hz sampling rate, a hit point sample is created after every 33.33 ms, limiting the temporal resolution of the gaze movement data available. Consequently, the available thresholds closest to the mode at 250 ms are 233 ms and 266 ms, which are equivalent to seven and eight samples respectively. We select the more conservative value of at least eight samples (~266 ms) as a threshold to identify data clusters containing fixations.

However, since the data still includes a considerable amount of missing data points, we expect a significant number of clusters to be "cut" by missing data points, thus appearing to be of shorter duration. Consequently, using a fixed temporal threshold to identify clusters containing fixations creates the problem of falsely failing to identify these "cut" clusters as clusters containing fixations. Therefore, to counteract this effect, it is crucial to interpolate missing data samples if possible.

Therefore, as a third step in our pre-processing algorithm, we interpolate short intervals of missing data. If fewer than eight consecutive data points (~266 ms) are missing, it is improbable that participants had enough time to make a saccade to a different spatial area (ROI), finish a fixation, and make a saccade back to the same area. This reasoning is done in close analogy to the reasoning of the threshold value to determine data clusters containing fixations above. Consequently, we interpolate data if fewer than eight consecutive data points are missing and only if these occur between two clusters on the same collider (ROI) (Fig 2A). In case these interpolation criteria apply, the interpolated data points are labelled with the collider name and combined with the data clusters surrounding the short interval of missing data. Larger gaps are not interpolated but treated as missing data. Furthermore, missing data points occurring between samples of different colliders are not interpolated, independently of the duration of the gap. This procedure ensures that most fixations are captured while minimizing false interpretations of missing data.

As a fourth step, we finally apply the temporal threshold to identify clusters that contain fixations during which the participants could process the visual input. As described above, we adopt a threshold of eight samples (266 ms) to identify clusters that contain at least one fixation (Fig 2B). Using this approach, shorter clusters likely to be caused by samples during saccades, i.e., periods during which perception was suppressed, would be excluded from further analysis. In the following, we will define the clusters containing fixations as new meaningful "gaze" events and they will form the basis of our further analysis.

This procedure classifies on average 86% of the data samples as belonging to gazes (Fig 2C). Previous studies reported that humans spent approximately 90% of viewing time on fixations

 

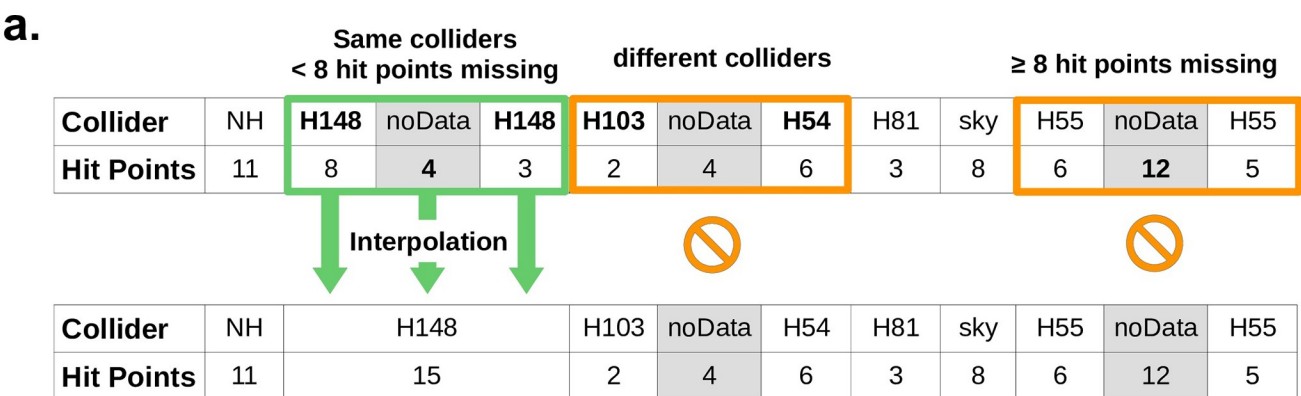

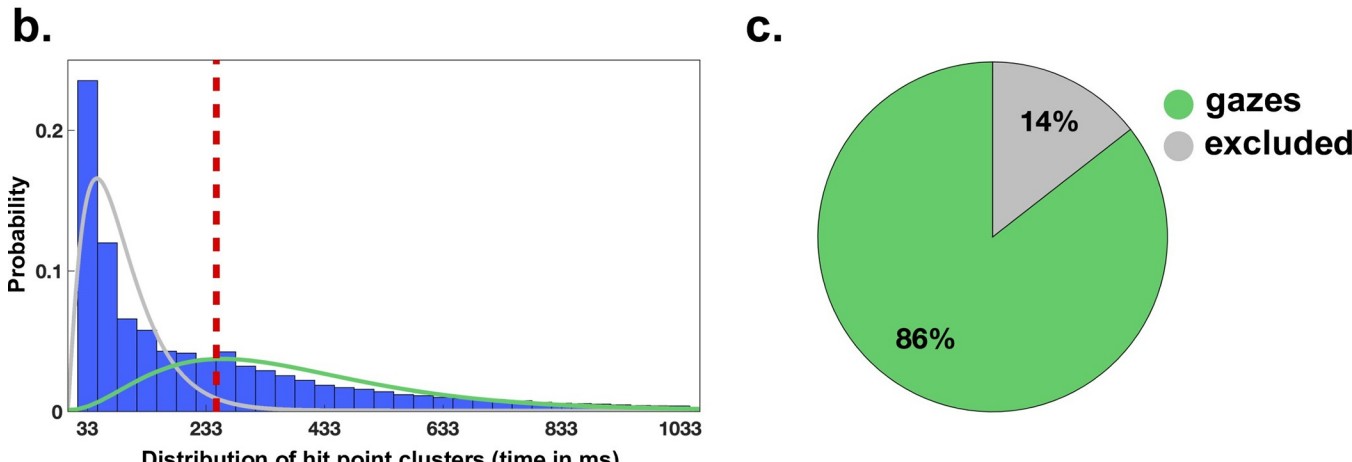

**Fig 2. Defining gazes. (a)** We interpolate missing data only if less than eight samples are missing consecutively (pupil detection with less than 50% probability) and if these samples occur between two clusters on the same collider. During the interpolation process, these samples are then unified with the two clusters to form a new cluster on the same collider. The first row shows three clusters of missing samples (marked noData), while the second row represents the result of the algorithm. In the first cluster (green box), both interpolation conditions apply: there are fewer than eight consecutive missing data samples (#4) and they are surrounded by two clusters on the same collider. Consequently, these missing samples are interpolated and combined to a new cluster. In the second cluster (1st orange box) the first interpolation condition applies (#noData samples < 8) but the cluster occurs between clusters on two different colliders (H103 and H54). Therefore, no interpolation is performed. In the third cluster (2nd orange box) only the second interpolation condition applies. Even though the missing data samples occur between two clusters on the same collider (H55), the first interpolation condition is violated (#noData samples ≥ 8). Consequently, no interpolation is performed. **(b)** Histogram of hit point cluster length distribution after interpolation. The distribution is visualized between 0 and 1033 ms. The longest hit point cluster on a house has a duration of 18.9 seconds. The ordinate corresponds to the probability of each duration. Since previous work used gamma distributions to model the distribution of fixation durations or response latencies [39,40], we model the two partly overlapping gamma distributions for visualization only, fitting the distributions of the duration of fixations (green) and non-fixation events (grey). The dashed red line marks the separation threshold for gazes. **(c)** The pie chart shows the result of the gaze classification across all participants.

[37]. This indicates that our attempt to capture gaze events under dynamic conditions in VR is on the conservative side.

## How to create graphs from gazes

To capture information gathered by the participants during exploration of the virtual town, we create gaze graphs based on the gaze events. In these graphs each node represents a house. Viewing two houses in the VR environment in direct succession gives information on their relative spatial location. Therefore, if anytime during the experiment a gaze event on a house is directly followed by a gaze event on another house the respective nodes are connected by an

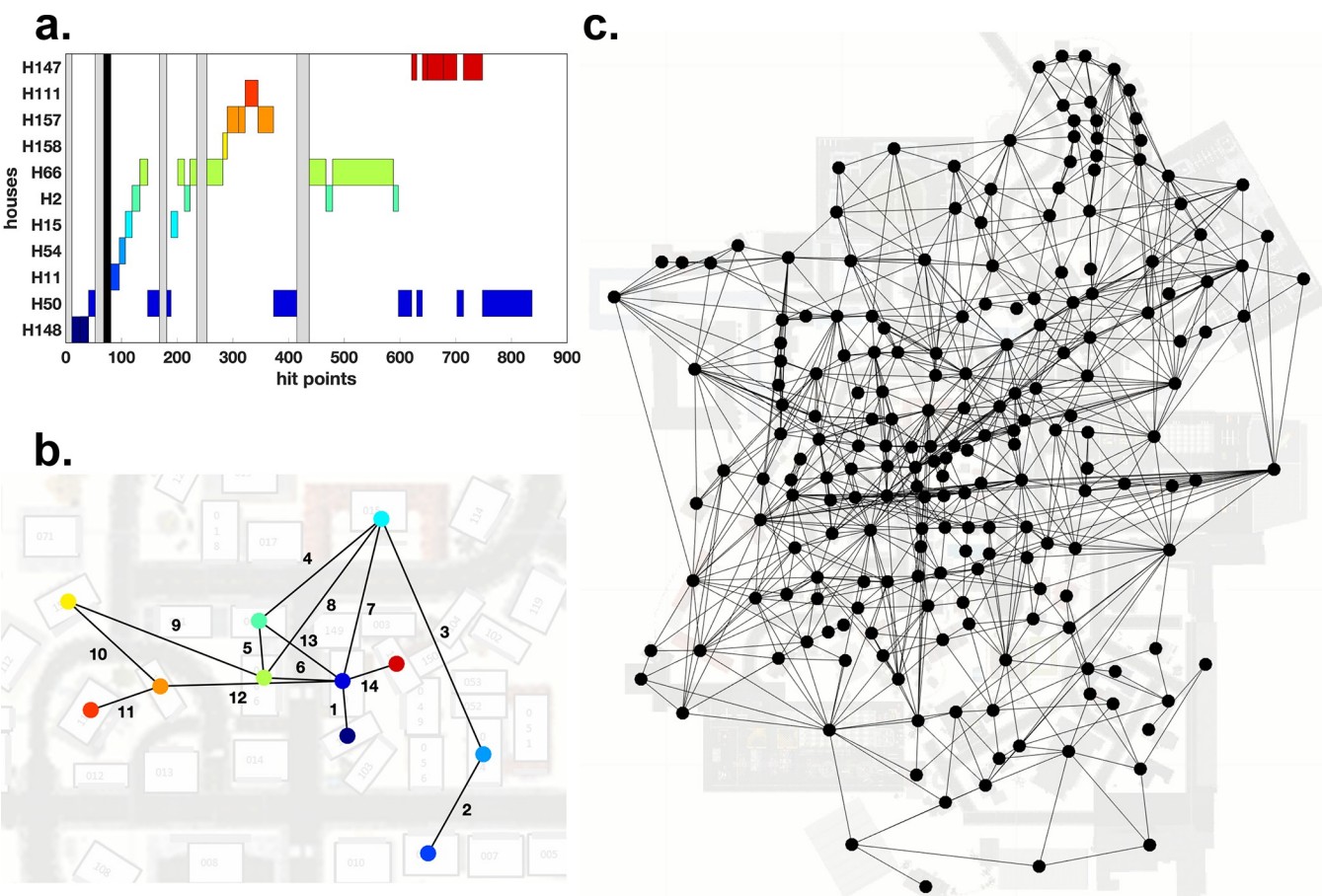

**Fig 3. Graph creation. (a)** Timeline of gaze events by a participant. The abscissa represents the first 30 seconds (900 hit points) of the recordings. The ordinate contains all viewed houses viewed during that time line. We number houses and name them accordingly, e.g., H148 for house number 148. In this panel each house has a distinct color for visualization only. The grey bars represent clusters of the NH category, which are not considered during graph creation. The black bar identifies a remaining cluster of missing data samples. Therefore, no edge will be created at this moment in the timeline. **(b)** The graph corresponding to the timeline of panel A is visualized on top of the map of Seahaven. The colors of the nodes match the colors of the boxes in panel A. Edges are labelled according to the order they were created. **(c)** The complete graph of a single participant based on all gaze events during 90 min of exploration visualized on top of the 2D map of Seahaven. Note that in this visualization the locations of the nodes correspond to the locations of the respective houses they represent in Seahaven, however, this locational information is not contained in the graph itself.

undirected edge. All gaze graphs are undirected and unweighted. Hence, they do not contain information about the directionality or frequency of their edges (i.e., direction and frequency of succession of gaze events). The completed gaze graphs capture the spatial information obtained by the visual exploration in the virtual town.

The first 30 seconds of one participant exploring Seahaven serve as an example of the process to create a graph (Fig 3A). The graph contains one node for each house (ROI), but does not consider the spatially unspecific no-house and sky category. The graph creation process starts with an edge between the first house viewed and the second house viewed (Fig 3B). If the gaze on a house is followed by a missing data cluster, then no edge is created. As stated above, the edges are unweighted, i.e., binary. That is, if the participant looked back to two houses already seen in sequence, an edge between these houses is already in place and the graph is not changed. The process of edge creation is iterated for all gaze transitions. Whenever a new house is viewed, a new node is created in the graph. Fig 3C shows a visualization of the result of applying this procedure to the complete data of the example participant.

The construction of the graphs uses two free parameters. The first parameter determines the width of gaps that are not interpolated. The second parameter sets a threshold for the minimal number of samples in a cluster required to define a gaze event. Both parameters relate to the typical length of a fixation and are set to a value of eight samples (k = 8). In order to estimate the impact of this choice, we rerun the graph generation algorithm with a lower (k = 7) and a higher (k = 9) setting. Averaged over participants, we obtain a mean number of nodes of 209.05+/-2.21, 208.20+/-2.65, and 207.55+/-2.78 mean+/-standard deviation for k = 7, 8, 9 respectively. Thus, the differences in the mean number of nodes for different parameter values are of the order of 1% of the total number of nodes. Averaged over participants the mean number of edges results in 898.1+/-139.03, 883.45+/-134.16, and 855.45+/-127.56 mean+/-standard deviation for k = 7, 8, 9 respectively. Thus, the differences in the mean number of edges for different parameter values are of the order of 2.5% of the mean number of edges. In this analysis the nodes and edges need not necessarily be identical, and the variance of the number of edges is large. Nevertheless, the number of nodes and edges is rather consistent comparing the mean number of nodes and edges over all participants of all three thresholds. The choice of the parameter does not appear to be critical and in case of doubt, k = 7 implies smaller changes than a choice of k = 9.

## Is it a single city or multiple suburbs? Graph partitioning

To address questions on spatial cognition, we are interested whether the virtual city should be treated as a loosely connected set of suburbs or as a coherent single city. The search for distinct clusters in the graph directly relates to the problem of graph partitioning. In the field of graph theory, partitioning is a well investigated problem as it divides a graph into smaller mutually non-overlapping subgraphs. In other words, the procedure of partitioning a graph has the objective to investigate whether the graph can be easily cut into two or more completely separate graphs. Here, "cut" is used as a term to remove an edge from a graph. For this, the number of connections between nodes is relevant. Nodes that form a cluster have a high number of within-cluster connections. Separate clusters are only connected with a few between-cluster connections. The fewer these between-cluster connections are the fewer need to be cut to separate the clusters from each other. Of specific interest are therefore approaches that maximize within-cluster connections (those connections are maintained) and minimize between-cluster connections (those are cut). If two distinct clusters would be found, the viewing behavior in Seahaven would be concentrated, in other words 'clustered', in two distinct areas and the city could not be treated as one coherent city. In that case, the following analyses would have to be applied to the separate clusters individually, ignoring and cutting the edges between them. Thus, graph partitioning is an important step to classify Seahaven as a set of suburbs or as one coherent city.

Graph partitioning, the search for clusters of nodes heavily connected within but few connections to the remaining nodes forms a difficult computational problem. In principle, it can be addressed by exhaustive search. However, even for graphs of moderate size, the computational requirements are staggering. Therefore, we employ the spectral graph analysis, which is a highly efficient procedure in graph theory. In the next paragraph, we give the technical details to allow exact reproduction. A more detailed definition as well as simple introductory examples are provided in the section Graph Theory–a detailed explanation of used graph measures. However, these details of "how" to obtain the partitioning of a graph into subgraphs are not critical for the further understanding of the results. The question "does such a partitioning have few or many connections between the subgraphs" is more relevant as described further down.

The spectral graph analysis is based on the eigenvalue spectrum of a network's Laplacian matrix that gives insights into the graph's connectivity [41,42]. First, we consider the adjacency matrix and the degree matrix of a graph. The former is a binary square matrix where each entry indicates whether an edge connects the two nodes / houses. The latter is a square matrix that contains the degree of each node on its diagonal and is zero otherwise. We calculate the Laplacian matrix by subtracting the degree from the adjacency matrix. By construction of the Laplacian matrix the smallest eigenvalue of this matrix is always zero. The spectral graph analysis then uses the $2^{nd}$ smallest eigenvalue of the Laplacian matrix. The larger the $2^{nd}$ smallest eigenvalue of the Laplacian matrix is, the higher is the overall connectivity of the network ($\mu_2$ = 0.300+/-0.097). The eigenvector of this $2^{nd}$ smallest eigenvalue is also called Fiedler vector [43,44]. This Fiedler vector has one entry for each node and gives insight into groups of nodes that are mostly interconnected with each other. The Fiedler vector (Fig 4C) has a negative and positive component that can be used to divide two clusters. The further the Fiedler vector is away from 0 the higher is the connectivity of the particular node within its cluster. Fig 4C shows that many entries of the Fiedler vector are around 0, meaning that those nodes are also connected with nodes from the other cluster, i.e., indicating the absence of separate clusters. A

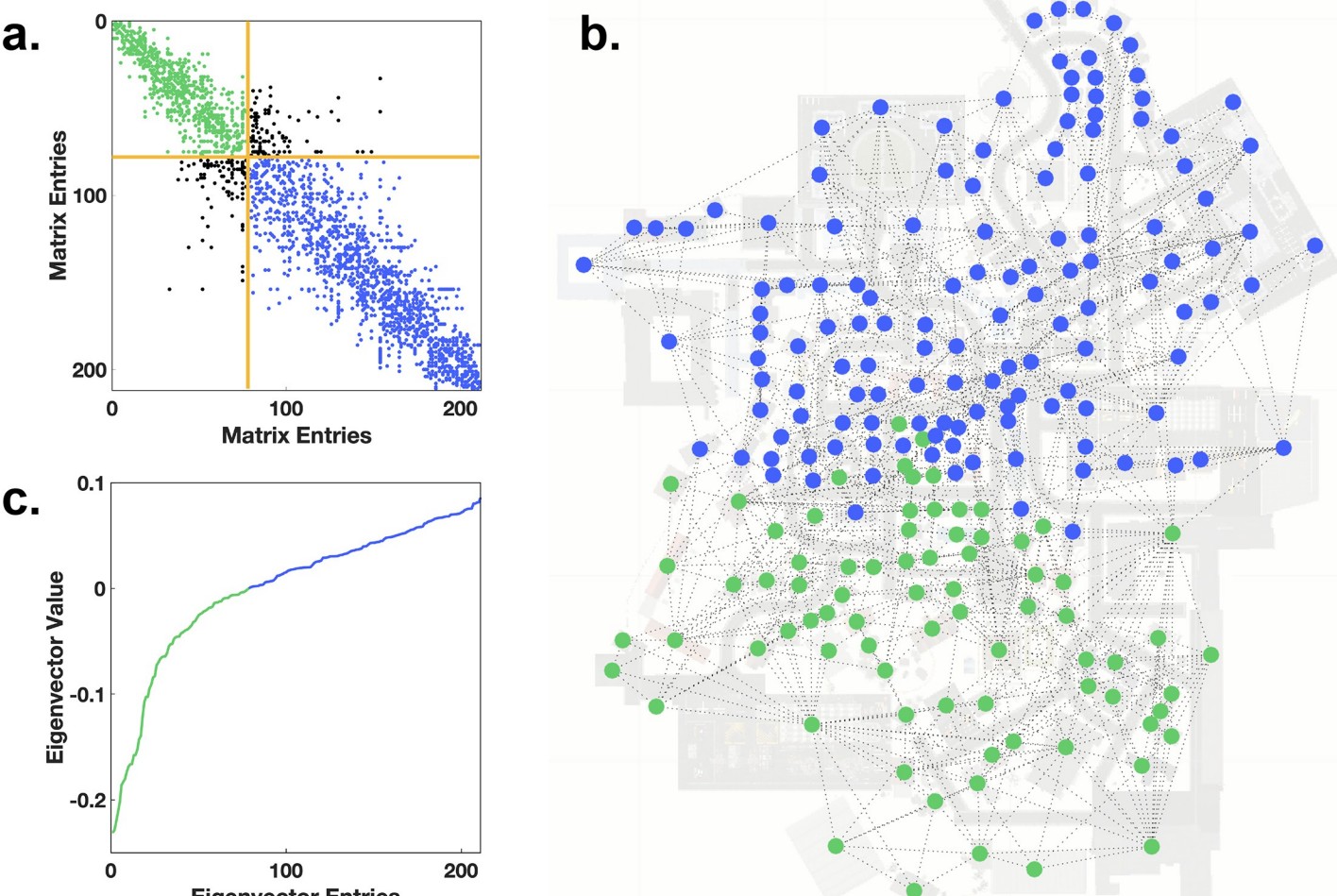

**Fig 4. Graph partitioning.** (a) The sparsity pattern of the graph's adjacency matrix sorted by the entries in second smallest eigenvector. Color coded into edges between nodes of one cluster (green), edges between nodes of the other cluster (blue), edges between nodes of the two clusters (black) and a distinction between the clusters (yellow). (b) The two clusters are displayed onto the map of participant 35. (c) The second smallest eigenvector of the Laplacian matrix is sorted ascendingly and color coded into two clusters.

Fiedler vector that indicates a clustering effect can be examined in the graph partitioning toy example described in the section Graph Theory–a detailed explanation of used graph measures. By sorting the adjacency matrix according to the sorting index of the Fiedler vector, the potential clusters can be visualized, which would manifest in a modular block pattern (Fig 4A). However, this pattern is not visible in the present analysis, illustrating the absence of distinct clusters, but can be also examined in the graph partitioning toy example in the section Graph Theory–a detailed explanation of used graph measures.

To investigate the quality of the partitioning, we follow the definitions used by Schaeffer [45]. As a measure of cluster goodness, we consider the intra-cluster density and the inter-cluster density. The density of a graph is defined as the ratio of instantiated edges relative to the number of possible edges:

$$\delta(G) = \frac{m}{\binom{n}{2}}$$

with $n = |V|$ as the number of nodes, and $m = |E|$ as the edge count. Furthermore, we define the intra-cluster density as the average of the densities of both clusters and the inter-cluster density as the ratio of inter-cluster edges to the maximum possible inter-cluster edges.

We analyze the graphs of individual participants based on 90 min exploration time. On average, the participants' graphs contain 883 edges. The mean density of the graphs is 0.041 +/-0.006. This means that compared to a fully connected graph (each house is connected to each other house) on average 4.1% of all possible edges are instantiated. Furthermore, dividing the graphs into two parts would require on average a cut of at least 9.3% of the edges of the graph. Specifically, a sufficient cut has to cut on average 82.6 edges resulting in two clusters with approximately 400 edges each. After partitioning, the mean intra-cluster density was 0.079+/-0.012, while the mean inter-cluster density was 0.0083+/-0.0020. These numbers indicate that the graph cannot be easily partitioned, without cutting a fair number of edges.

In conclusion, our results reveal that the graphs cannot be distinguished into large-scale clusters. That is, the exploration of Seahaven does not show separate city blocks, but was rather well-balanced. Thus, the virtual environment can be treated as one coherent city.

## The distribution of gazes on houses – Node degree centrality

We characterize the role of different houses during visual exploration by indices adapted from graph theory. The node degree centrality is the main and most basic graph-theoretical measure in graph-theoretical research. It is defined as the number of edges connecting a node, which is the sum over the entries of a column of the adjacency matrix:

$$c(i) = \sum_{j}^{N} x_{ij}$$

with $i$ being the node under consideration and $x_{ij}$ the adjacency matrix. Here, the node degree of a house reflects the number of other houses a participant made a gaze to or from the house under consideration in direct succession. Thus, the node degree centrality in the gaze graph differentiates houses according to their importance during visual exploration in the virtual town.

First, we use the node degree centrality to investigate the viewing behavior on an individual participant level. Fig 5A shows the gaze graph of a participant with individual nodes color coded according to the respective node degree. Whereas many nodes have a degree centrality in the single digit range, a few houses reach rather high values. The node degree for each participant ranged up to 33. It is apparent that the range of node degrees is surprisingly large.

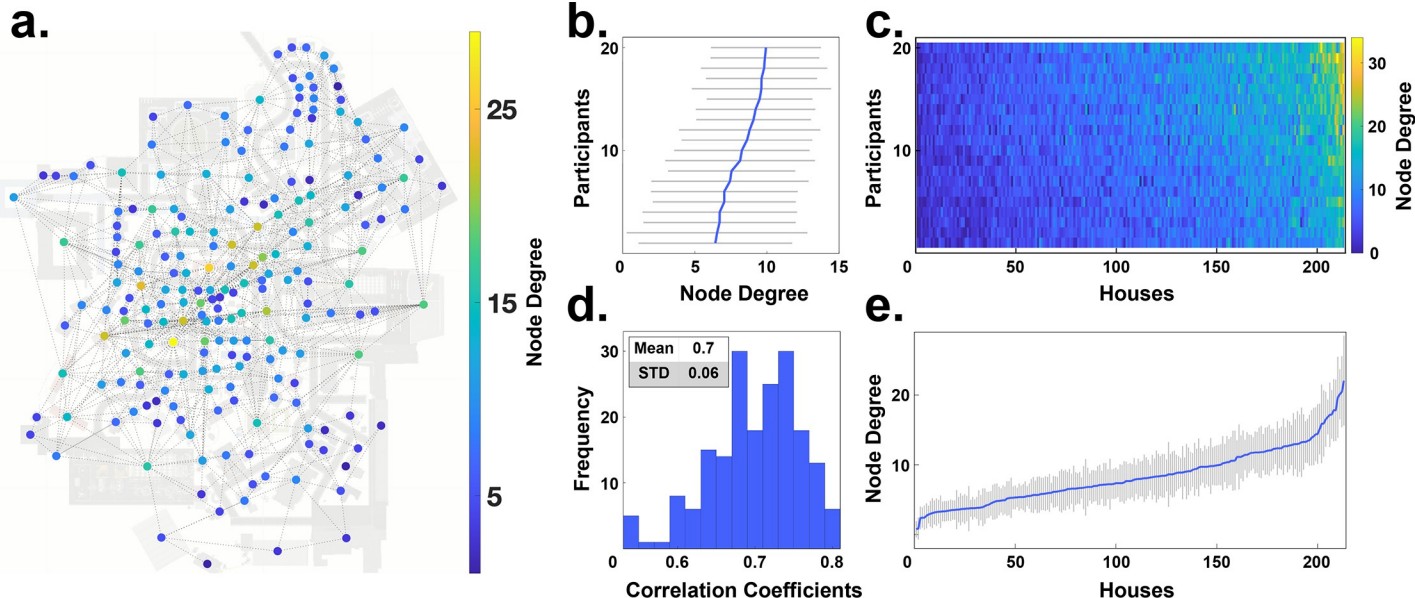

**Fig 5. Node degree centrality. (a)** The graph of one participant is visualized on top of the map of Seahaven. The nodes were colored according to their respective node degree centrality. **(b)** The individual mean node degrees of all participants (blue line) and their respective standard deviation (grey lines) are depicted sorted by increasing values. **(c)** A pseudo 3D plot color coding the node degree of every house (abscissa) for every participant (ordinate). The houses are sorted so that the average node degree value increases along the abscissa. Similarly, the participants are sorted, so that the average node degree increases along the ordinate. The marginals of this plot result in the panels b and e. **(d)** The distribution of the pairwise inter-participant correlation coefficients of the node degree values of all houses. **(e)** The mean node degree of each house sorted according to the mean node degree along the abscissa (blue line) and their respective standard deviations (grey lines).

Next, we investigate the similarity of the node degree centrality distribution over participants. We calculate the average node degree of each participant across all houses and for visualization sort the participants in increasing order (Fig 5B). This reveals limited variations between participants. The average node degree of each house across all participants (Fig 5E) showed a monotonous linear increase. Both aspects are also combined in a pseudo-3D visualization (Fig 5C) that matches the order of the houses and the order of participants. Sorting the houses according to the average node degree centrality leads to a near monotonous increase on the individual participant level as well. Thus, the set of houses with highest node degrees are largely overlapping between participants. This was quantified by the high correlation of 0.70 (+/- 0.06) of node degree centrality between participants (Fig 5D). In summary, the node degree centrality varies considerably between houses, while the values of individual houses are rather consistent across participants.

The distribution of node degree centrality over houses reveals interesting aspects. Over a large range of houses the average node degree centrality increases only slowly. Only for the last few houses, we observe a steep increase. These houses stand out from the other houses and are viewed directly before or directly after viewing many other houses. Therefore, the high node degree centrality houses may serve as important reference points.

In summary, the node degree is a simple yet powerful centrality measure that can be used to identify important nodes in the gaze graph, hence important houses in visual behavior. First results indicate that a small number of houses show an especially high node degree across all participants, setting themselves apart from the rest of the city. Interestingly, a small number of houses with participant independent high node degree values, i.e., houses with many visual connections to other houses, would also be the characteristics that we would expect landmarks to display in gaze graphs.

### High node degree centrality houses – Hierarchy index

To further investigate the houses with a high node degree in respect to their distribution, we apply the graph-theoretical measurement of the hierarchy index. This index characterizes hierarchical configurations within networks. By applying it to the degree values that are above the participant's respective median, we focus the index on the upper tail of the distribution of node degree centrality. Here, we fit in a bi-log plot of the node degree centrality with frequency against the node degree values with the slope of the regression line starting from the participant's respective median, depicted in Fig 6A for one example participant with a slope of -2.6349. Performing this analysis for all participants results in a slope smaller than -2.00 throughout, with a mean of -2.91. Furthermore, the small standard deviation of 0.34 showed, that the hierarchy index is similar across participants (Fig 6B). According to Rodrigue [46], networks with strong hierarchical configurations, i.e., with many low degree nodes and few high degree nodes, had a slope of below -2 (or above 2). Therefore, the hierarchy index reveals a clear hierarchical structure of the gaze graphs and emphasizes the importance of the few exceptionally high node degree centrality houses.

This observation triggers a detailed look at the mean node degree distribution over all participants to identify those special high node degree houses. Plotting the mean node degree of houses averaged across participants onto the map of Seahaven (Fig 7C) highlights the scattered subset of high node degree houses. Furthermore, the mean node degree across all houses and all participants measures 8.3 with a standard deviation of 3.98 (Fig 7A). We select the value of the 2-sigma distance (16.25) as the threshold to identify the high node degree houses. This results in a set of 10 houses in Seahaven with node degree centrality values exceeding the threshold.

Next, we analyze these houses with respect to their interconnectivity and visibility in the city. The distribution of the node degree centrality over participants for each of the 10 houses in Seahaven reveals considerable variance (Fig 7B). However, all 10 houses for all participants have a node degree centrality of 9 or above. Plotting them jointly with all their edges onto the map of Seahaven for our example participant shows that they are located centrally within the city. However, these houses are connected to the outer areas of the city and together cover nearly the whole city (Fig 7D). All in all, these 10 high node degree houses, are located centrally in the city and their connections reach out into outer areas, i.e., they are viewed from much of the city area.

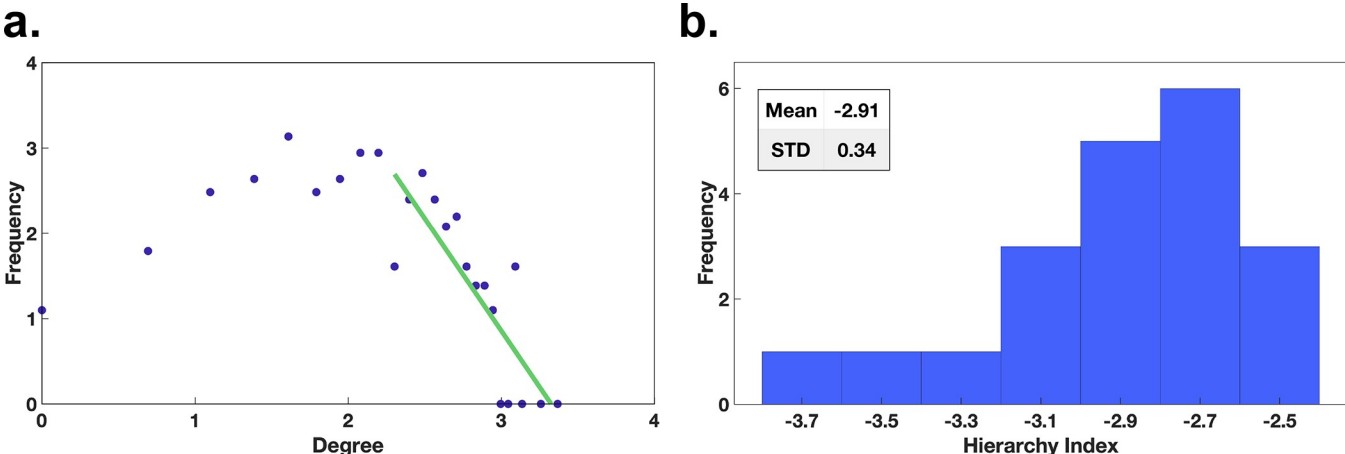

**Fig 6. Hierarchy index. (a)** The frequency of occurrence of the node degree for a single participant. The green line indicates the linear regression starting at the median of the distribution. **(b)** The distribution of the hierarchy index across all participants.

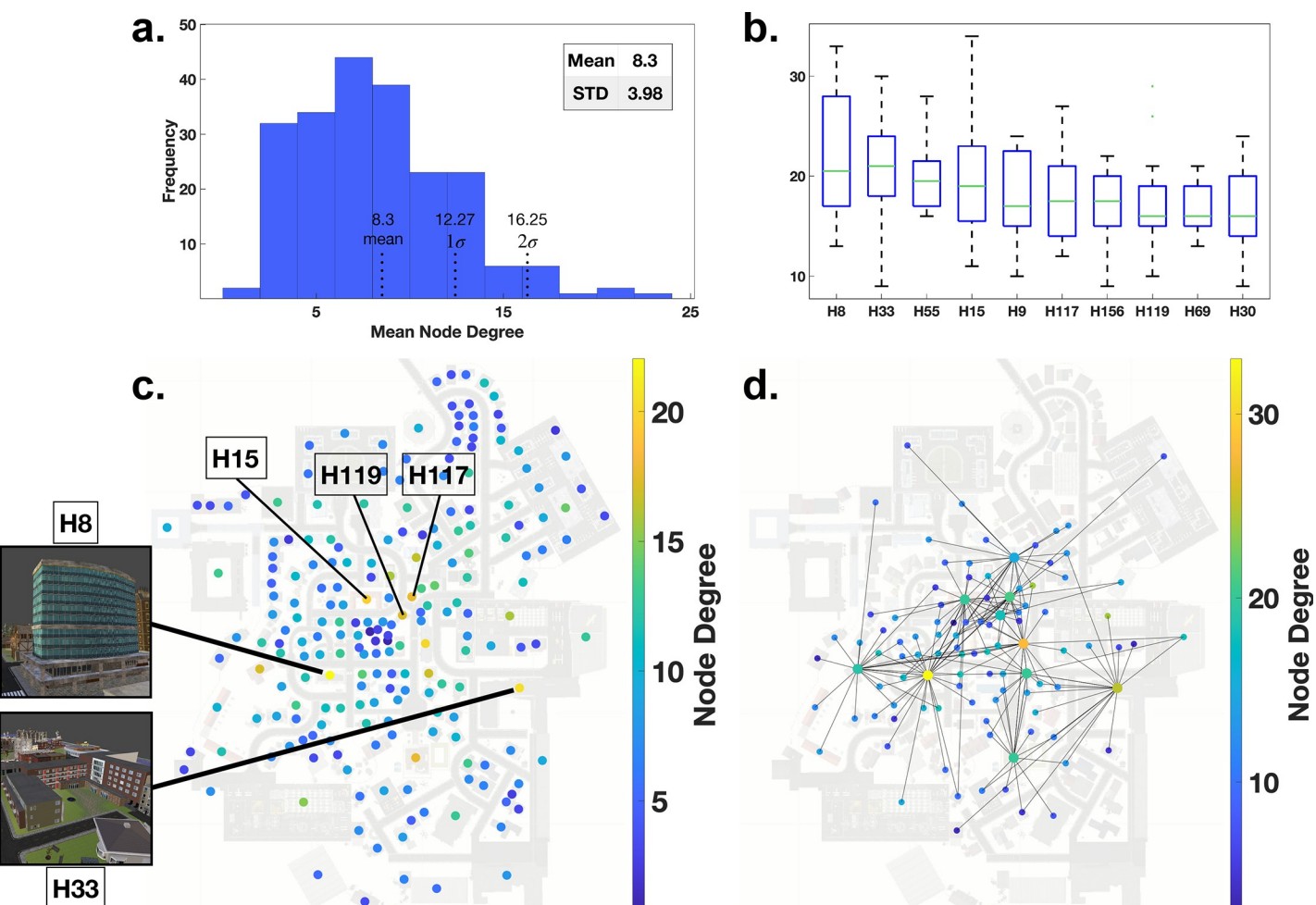

**Fig 7. The high node degree centrality houses. (a)** The mean node degree distribution across all participants with mean, 1σ- and 2σ-thresholds. **(b)** A box plot of the 10 houses with at least 2σ-distance to the mean node degree. **(c)** A map plot with all nodes color coded with their respective average node degree across all participants. **(d)** The 10 houses, which exceeded the 2σ-distance to the mean displayed on the map for our example participant with all their connections and color coded with their respective node degree.

The characteristics of high visibility of a small number and similar use over participants would be expected in landmarks. Our results show that the node degree distribution is similar across participants and that only a few houses have exceptionally high node degrees. This is supported by the hierarchical configuration of the network. The 10 houses with an average node degree distribution exceeding a 2-sigma threshold are more centrally located and had viewing connections into the outer areas covering nearly the whole city. All things considered, these findings suggest the notion that this set of houses is exceptional across multiple domains and displays the characteristics expected from landmarks. Therefore, in the following we will refer to these 10 buildings as "gaze-graph-defined landmarks".

## The connections between the gaze-graph-defined landmarks – The rich club coefficient

In this section we investigate, whether the gaze-graph-defined landmarks serve as the core of a network that could be used for navigation in the city. For a quantitative investigation, we

applied the concept of the rich club coefficient to our gaze graphs. The rich club coefficient is a frequently practiced graph-theoretical method in network theory and was initially applied in internet network analyses [47]. Yet, the rich club coefficient has also been transferred to neuro-scientific contexts. The approach has been used to map out both subcortical and neocortical hub regions and to show that those regions with high linkages are also highly connected between each other and, thus, indeed form a rich club [48]. In this study, the rich club coefficient allows one to quantify in how far gaze-graph-defined landmarks are preferentially connected to each other.

We calculated the connectivity between a subset of nodes with at least a specific degree value using:

$$RC(k) = \frac{2E_{\geq k}}{N_{\geq k}(N_{\geq k} - 1)}$$

with $k$ as the set node degree threshold of the rich club, $E_{\geq k}$ as the number of edges between nodes with degree larger or equal to $k$, and $N_{\geq k}$ as the number of nodes with degree larger or equal to $k$. Thus, the rich club coefficient is the fraction of edges instantiated between nodes of degree $k$ or larger and the total number of edges possible between nodes with the same degree or larger. In other words, we first check how interconnected the whole graph is and subsequently decrease the subset of nodes considered by removing low degree nodes from the set.

For the interpretation of the rich club coefficient, we need a baseline. For that purpose, we compare generated random graphs with similar statistics of node degrees and calculate their rich club coefficient. For each participant, we generate 1000 random graphs with the same number of nodes as the respective original graph. Of these, we select the 10 graphs with the most similar, in terms of the two-sample Kolmogorov Smirnov test, distributions of node degrees to the original distribution and use these as a baseline. Subsequently we take the ratio of the original rich club coefficient and the coefficient of one of the control random graphs. Finally, we average this ratio over all control graphs. This results in an adjusted rich club coefficient that compares the connectivity between those houses that pass the threshold with a random graph with matched node degree statistics. A value of 1.0 indicates no bias in connectivity, higher values would indicate a preferential connectivity within the considered set of nodes.

As a next step we investigate the rich club coefficient as a function of the threshold node degree. For example, for a threshold of 1 all nodes of a degree 1 or larger are considered. For this case we observe a coefficient at 1.0 (Fig 8A). Raising the threshold to 2, i.e., excluding all nodes of degree 1, leads only to a minor increase of the index. That is, houses of degree 2 or higher are preferentially connected within this group only to a small degree. This indicates that such a liberal threshold including the larger part of all houses does not reveal a meaningful bias in graph connectivity. Increasing the threshold value, i.e., removing low degree nodes from the set of considered nodes for the connectivity calculation, results to a monotonically increasing coefficient rising up to 1.5 (Fig 8A). Thus, the subset of high degree nodes is preferably interconnected in comparison to a random graph. However, increasing the threshold node degree naturally leads to a smaller number of nodes included, hence, increasing variance in the estimate. Therefore, we cut off the plot at a threshold value of 13, which is the mean node degree centrality plus one standard deviation rounded up (compare Fig 7A). Please note that the rich club coefficient is a continuous value. That is, there is no single threshold value defining a strict qualitative boundary. Instead, the selectivity of connectivity rises continuously. We consider the threshold value of 13 as the value with the largest bias, where the index can be reliably determined. This demonstrates that the nodes with a degree of 13 and above,

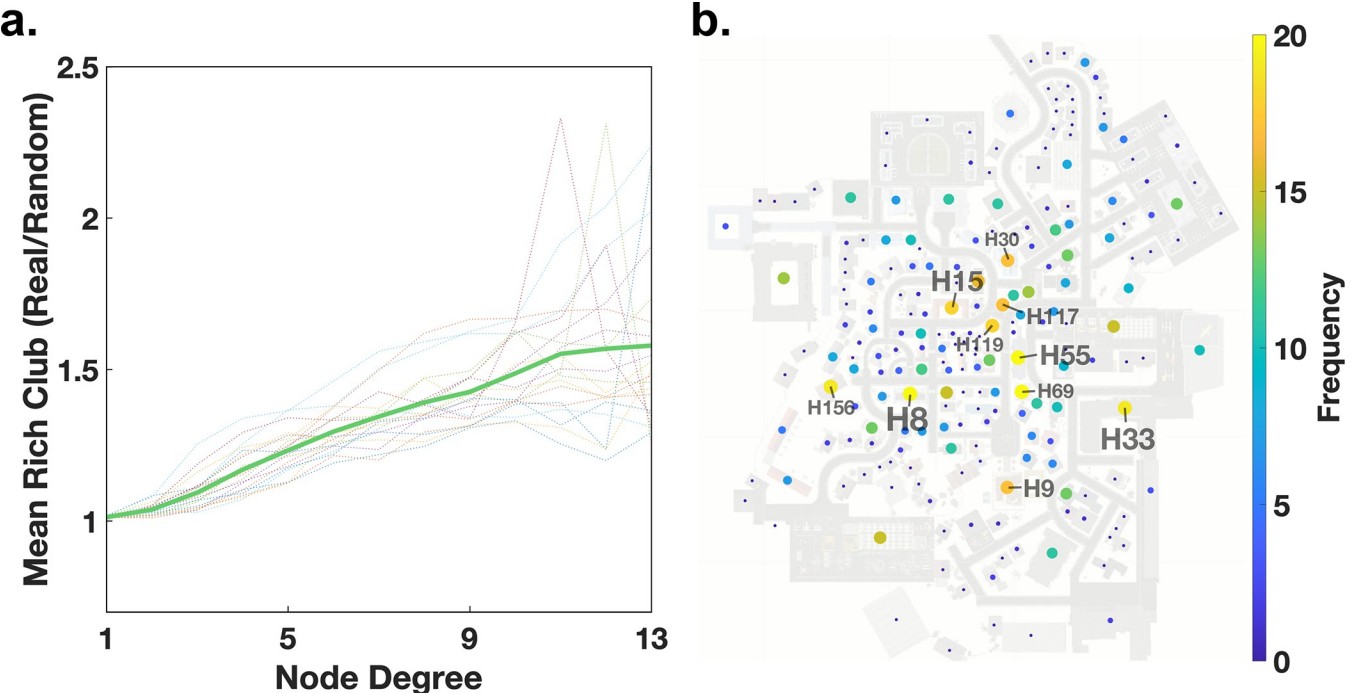

**Fig 8. The rich club coefficient. (a)** The development of the rich club coefficient with increasing node degree. The dot-lines are the rich club coefficients of individual participants, while the green line is the mean across all participants. **(b)** All houses displayed on the map both color coded and size coded according to their frequency of being part of the rich club across participants.

are interconnected much more than expected by chance and can therefore be considered as part of a rich club.

As a final step, we count the nodes with a node degree of at least 13 for each participant subsequently creating a list of houses and their frequency of having a degree of at least 13. From that list, we identify all houses with a frequency of being part of the rich club that exceeds the two-sigma distance from the mean frequency of all houses resulting in a selection of 14 houses. These 14 houses have the highest frequency of being part in the rich club across all participants and therefore have the highest interconnectivity on average. Furthermore, when comparing the 14 houses with the highest interconnectivity and the houses with the highest node degree, we find that the subset of the top 10 houses out of the selection of 14 high frequency rich club houses are the same houses that were identified as the top 10 node degree houses, i.e., gaze-graph-defined landmarks, earlier (Fig 8B). This connection is expected since the rich club is based on a node degree threshold and the node degree distributions of participants have a very high correlation (Fig 5D) meaning that the rich club nodes have high node degrees across participants. Overall, this analysis does not only confirm that the high node degree nodes are participant independent, but it gives evidence for a highly interconnected network of gaze-graph-defined landmarks in the city, a rich club.

## Spatial arrangement of the gaze-graph-defined andmarks – Triangulation

To elucidate the role of the gaze-graph-defined landmarks in spatial navigation, we explore whether they could serve as a basis for triangulation. Triangulation is a method to infer one's own location based on the viewing angle in respect to two location anchors. Our analysis has revealed that the gaze-graph-defined landmarks form a highly interconnected rich club. Thus,

**a.**

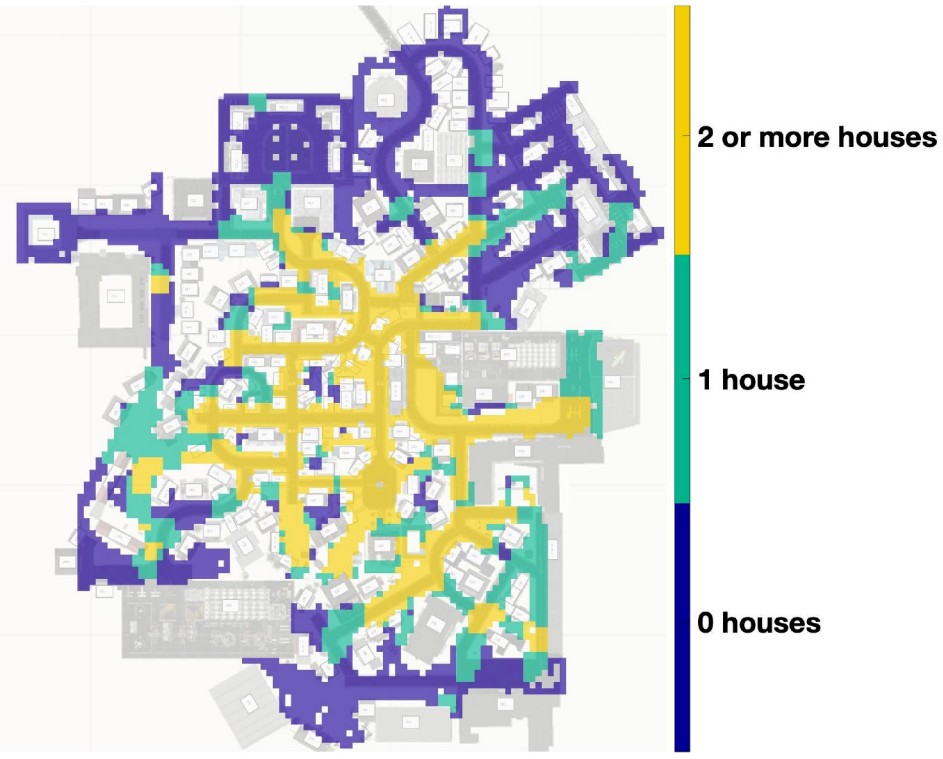

**Fig 9. Triangulation.** Location data of all participants plotted on the map of Seahaven. The color code indicates how many of the gaze-graph-defined landmark houses were viewed by participants at each location.

if the gaze-graph-defined landmarks are visible from most places in the city they could serve as a basis for triangulation.

As a first step we determine the parts of the city where at least one or two of the gaze-graph-defined landmarks were viewed by the participants. We evaluate the spatial distribution of our participants during exploration of the virtual town and how many gaze-graph-defined landmarks were actually viewed from each location. This analysis is performed with a spatial resolution of 4x4 m and an additional smoothing with a 3x3 m kernel. Please note that this analysis depends on the actual gaze data and, thus, reflects from where participants actually did view the gaze-graph-defined landmarks. Next, we differentiated the locations in three categories: zero/one/two viewed gaze-graph-defined landmarks. The resulting map represents the potential of triangulation based on the gaze-graph-defined landmarks at different locations in the virtual city (Fig 9). In 39.1% of the city areas that were visited by participants, two or more gaze-graph-defined landmarks were viewed from that location, potentially providing the basis for triangulation. In an additional 32.7% of the city area, exactly one gaze-graph-defined landmark was viewed. Only in 28.1% of the visited city area, none of the gaze-graph-defined landmarks were viewed. Weighting the city areas with the absolute time of presence in respective city areas, participants spend an even bigger fraction of the experiment time in areas where the theoretical basis of triangulation was given. Specifically, participants spent 53.2% of the experiment time in areas where at least two or more gaze-graph-defined landmarks were viewed, 19.4% of the time in areas where one gaze-graph-defined landmark was viewed and 27.4% of the time in areas where none was viewed. The latter regions were mostly at the fringes of the city map. Our data does not allow any conclusion on whether the participants actually

performed triangulation or whether this option influenced their exploration. However, in principle triangulation based on gaze-graph-defined landmarks is possible in areas of the city where participants spent slightly more than half of their time.

## Discussion

In this study, we establish a method to quantify characteristics of visual behavior by using graph-theoretical measures to abstract eye tracking data recorded in a 3D virtual urban environment. We define gazes of participants that freely explored the virtual city Seahaven, and use these to convert the viewing behavior into graphs. In these gaze graphs, nodes represent houses while edges represent their visual connection, i.e., gazes in direct succession on the respective houses. Thus, the gaze graphs capture relevant spatial information gathered during exploration. The node degree centrality graph measure reveals a surprisingly large variance. However, the values of individual houses were rather consistent across participants, as shown by the high mean correlation. Additionally, we observed that the degree distribution across houses increased steadily, except for only a few high node degree houses. These houses may serve as important reference points, the so-called gaze-graph-defined landmarks. The analysis of the hierarchy index demonstrates that the frequency of houses decreased drastically with increasing node degree, revealing a hierarchical graph structure. The set of identified gaze-graph-defined landmarks were indeed preferentially connected, as demonstrated by the rich club coefficient. Finally, participants spent more than half their exploration time at locations, where at least two of the houses of the rich club were viewed, allowing in principle triangulation for spatial navigation. Thus, we presented a graph-theoretical approach for analyzing gaze movements supporting spatial cognition.

### The importance of the VR setup for spatial cognition research

The process of spatial navigation is abundant in everyday life, therefore investigating the process of gathering spatial navigation under natural conditions is our goal. New mobile eye tracking systems allow investigating spatial navigation and the role of landmarks under more natural conditions in real-world environments while recording real-time eye tracking data, thus capturing the most important sense for spatial navigation [33]. For example, Ohm et al. [22] investigated the selection process of landmarks in large scale indoor environments via the visual attraction measured with mobile eye tracking and evaluated pedestrian navigation systems based on indoor landmarks [23]. Furthermore, Wenczel et al. [27] found differences in gaze behavior during incidental and intentional route learning in a natural outdoor environment with intentional learning leading to more fixations on landmarks. However, studies conducted with mobile eye tracking systems in natural environments are usually challenged with several disadvantages. In addition to the poor accuracy of mobile eye tracking systems, another major issue comes with identifying frames that contain areas of interest during the data pre-processing. Often, identification of regions of interests and therefore identifying the relevant fractions of the eye tracking data could only be solved by manual detection in each frame [22,26,27]. Furthermore, the natural environment allows only limited control of the experimental conditions, especially regarding variances in light affecting the eye tracker systems [26], and variance in the environment due to other people or traffic [24,25]. Depending on the system, eye tracking calibration can be distorted for some distances, therefore limiting the valid distance of gazed objects that can be analyzed [26]. Consequently, most eye tracking studies struggle to be conducted under natural conditions and thus reduce the ecological validity of gathered information of viewing behavior under natural conditions.

In the present study we follow the idea that virtual reality allows experimental paradigms in VR with high control while still providing a more naturalistic environment. However, given that spatial navigation is based on multimodal sensory input, it can be questioned whether a virtual reality setup to investigate spatial navigation is ecological valid. Specifically, in the present study we employ a virtual environment that is more restricted in sensory input than the real world, i.e., limited to visual and vestibular input information. In contrast, recent literature highlights the importance of idiothetic and sensorimotor information about self-position and about self-motion [49,50]. Nevertheless, the dominating sense of spatial perception is vision [33]. Only this sense can gather reliable information of space and the environment independently of the physical distance and also allows one to perceive topographic characteristics over large distances. Therefore, by observing visual attention and visual behavior using the method of eye tracking, we could gather important insights about the usage of spatial cues, that indicate cognitive processes related to spatial navigation, specifically landmark usage. Moreover, using a VR setup with additional eye tracking providing a solution to the technical problems observed in previous studies in real-world environments. Furthermore, our proposed analysis method using hit point clusters and gaze events allows a fast and precise analysis option, including the automated identification of ROIs and a cleaning and interpolation pipeline. Therefore, the proposed analysis improves the ecological validity of the analyzed data by considering the reduced sampling rate, increased noise sources, and extended set of eye movements expected in a 3D environment compared to common 2D desktop eye tracking setups. Consequently, applying the experimental paradigms using the VR setup and eye tracking analysis proposed in this paper, provides an ecological valid option to investigate spatial navigation under naturalistic conditions, while avoiding the problems observed in real-world environments.

## Accuracy of eye tracking data recorded in virtual reality

Resting our analysis on eye tracking data allows us to observe and investigate participants visual attention during spatial navigation processing. However, the combination of eye tracking and a head mounted virtual reality headset comes with several challenges. Accuracy of mobile eye tracking systems is often reduced compared to other systems [51]. Specifically, in VR experiments, this is often due to the freedom of head movements and weight of the VR headset. Since typical errors due to slippage and head movements increase over time, we conducted a short validation and if necessary, a complete calibration and validation procedure every 5 minutes during the experiment [34]. Nevertheless, the mean validation error of 1.55˚ before experiment start and 1.88˚ during the experiment is rather high compared to classical lab-based eye tracking studies. However, unlike lab-based eye tracking studies, our preprocessing and analysis is based on hit point clusters that fell on the same collider in the VR environment. Thus, we summarized data points that were located in rather close proximity. Though the notion of close proximity must be taken with care since most colliders still corresponded to the size of a complete house, therefore making a small deviation in gaze location due to the validation error less problematic. All in all, considering that the preprocessing is based on the spatial distribution of the accumulated hit point clusters, a minor increase of validation error should not affect our data significantly.

## The data type of "gaze events"

One major factor in the analysis of eye tracking data recorded in an VR environment is the algorithm differentiating the different types of eye movements and in our case the definition and creation of the data form "gaze". Essentially, the four different types of eye movements

expected to occur in such a natural setting, i.e., saccades, fixations, vestibulo-ocular reflexes, and smooth pursuit movements, can be separated into two categories. On the one hand, we have fixations as the typical source of visual perception. Since vestibulo-ocular reflexes and smooth pursuit movements stabilize the retinal image in dynamic situations, they lead to visual perception similar to the input during fixations [37]. Thus, we can classify fixations, vestibulo-ocular reflexes, and smooth pursuit movements as the first category of visual perception. Saccades, on the other hand, render the participant blind to the momentary visual input, hence a hit point sample created during a saccade will not have been perceived by the participant [37,38]. Consequently, it is essential to differentiate the data between the first category of visual perception and saccades. Usually, this is done either by velocity or gaze location-based algorithm. However, the virtual environment in Unity3D results in three impeding factors. Firstly, the complexity of the VR environment in Unity3D only allowed for a 30 Hz sampling rate, thus it does not allow a saccade detection based on sudden changes in gaze velocity. Secondly, the estimation of gaze location based on the ray cast process is limited to whole colliders, often covering the size of a house, therefore it does not allow to identify small changes in gaze location. And thirdly, depth perception is usually not accounted for in saccade detection algorithms, therefore making it questionable to apply to eye tracking data recorded in 3D. Consequently, the available eye tracking data recorded in the VR environment did not provide the reliable information necessary in classical fixation detection algorithms that allows one to cleanly separate the different eye movements. Therefore, defining the new data form "gaze" allows for a functional method to clean the data. This process did not allow the identification of single eye movements nor did it exclude all saccades from the data. However, by identifying the data clusters that contained at least one fixation and excluded the data clusters that very likely did not contain any fixation, we could clean the data from samples that were unlikely to be visually processed by the participant. This is further supported by our data, since the process identified 86% of the data as gazes on average across all participants. With approximately 90% of viewing time being expected to be spend on fixations [37], the process identifying gazes appears to be on the conservative side. Overall, with higher sampling rates available in virtual environments, new options to identify saccades might become available. However, we believe that given the data we have available, defining the new data form "gaze" was the best option to differentiate between stray samples unprocessed by the participant and meaningful data carrying important information of the first category of visual perception including fixations, vestibulo-ocular reflexes, and smooth pursuit movements.

## Gaze graphs

The graph-theoretical analysis approach used in this study is crucial, since it leads to a large compression and abstraction of the data. Using this method, a few million gaze samples were condensed into a few graphs. In general, graph theory is used in many areas to make complex information of pairwise relations accessible [52–54]. This process includes several decisions, which might critically influence the later analysis. First, the visibility of the houses depended on the participant's location in the virtual world, which is not necessarily close to one of the respective houses. Accordingly, the participant might view two houses in direct succession leading to a visual connection, even though the houses themselves might not be visible from each other's location. Second, in case the data contained a cluster of missing data points no information about visual connections during this time was available and no edge could be created between nodes. Third, the gaze graph was undirected and unweighted. It did not contain any information about the order in which the two successive gaze events took place. Consequently, the gaze graphs only contained information about whether a visual connection took

place at some point during the experiment, and not how often. It is easy to conceive that quantitative information on the direction and frequency of gaze trajectories contains useful information. Further, graph theory provides a variety of tools applicable to such enriched information. While we do consider further studies with enriched graphs to be a promising avenue for future studies, for the present first step we deliberately kept the attempt to utilize gaze-graphs for research of spatial cognition simple. Applying the above considerations for graph creation, our graphs represented the gaze data in a well-defined and meaningful way.

## Gaze graphs and landmarks

Our results of the graph-theoretical analysis revealed a small subgroup of houses that matched several characteristics of landmarks. Landmark knowledge refers to the knowledge of salient objects that serve as orientation anchors and are memorized when exploring a new environment [55–57]. Since the node degree is a common measure to investigate the importance of single nodes in the network [58], we expected landmarks to stand out in visual behavior compared to other houses and therefore show high importance in the gaze graph visible in high values of their node degree centrality. The gaze graph represented the visual connections between houses, therefore, the node degree centrality measured how many different houses were visually connected with the house in question. If a house would serve as a landmark, we would expect participants to often create visual connections between the landmark house and other houses. The results of our node degree centrality analysis showed a clear difference between a subgroup of houses with the average node degree values of the houses exceeding the mean node degree over two times the standard deviation. Moreover, those houses were above chance level interconnected with each other forming a rich club. The distinctiveness of these buildings is strengthened by the high mean hierarchy index. Taking all these findings into account, we suggest that these houses fit the characteristics a landmark would display in a gaze graph and thus propose to call this subgroup of houses gaze-graph-defined landmarks.

This interpretation of our results was supported by further characteristics of the gaze-graph-defined landmarks. In a navigational context, the term landmark usually refers to any type of object that is highly visible or easily recognizable in the environment and thus serves as a point of reference [2,59]. In addition, landmarks are often differentiated into global and local landmarks [3]. Typically, global landmarks can be seen from far distances and provide a reference for beacon and directional and more compass-like orientation, whereas local landmarks might only be visible in a local area and are often located at road crossings developing action-relevant spatial information [3,8,9]. Looking at the visual appearance of the gaze-graph-defined landmarks, we found typical characteristics of landmarks including visual saliency due to size, color or location in the city with a high variability in appearance between the defined landmarks. For example, the house with the highest average node degree across all participants is higher than most surrounding buildings and has a large distinct blue window front that sets it aside from its surrounding, thus suggesting its function as a global landmark (Figs 6B, 7B and 10A1). Additionally, it is located in the very center of the city making it visible from most parts of the city. The house with the second highest average node degree across all participants stands out due to its size regarding its surface area but not its height (Fig 10A2). It is located next to the main road that connects the north and south part of the city, even though it is only visibly in the south-east part of the city. This location at an important crossing in the city for navigating might make it a local landmark with navigational relevance. In general, most of the gaze-graph-defined landmarks are located next to crossings of main roads in the city thus fitting a characteristic of local landmarks [3]. It has to be considered that our gaze-graph-defined landmarks have some inadvertent features even though we took care in designing the virtual

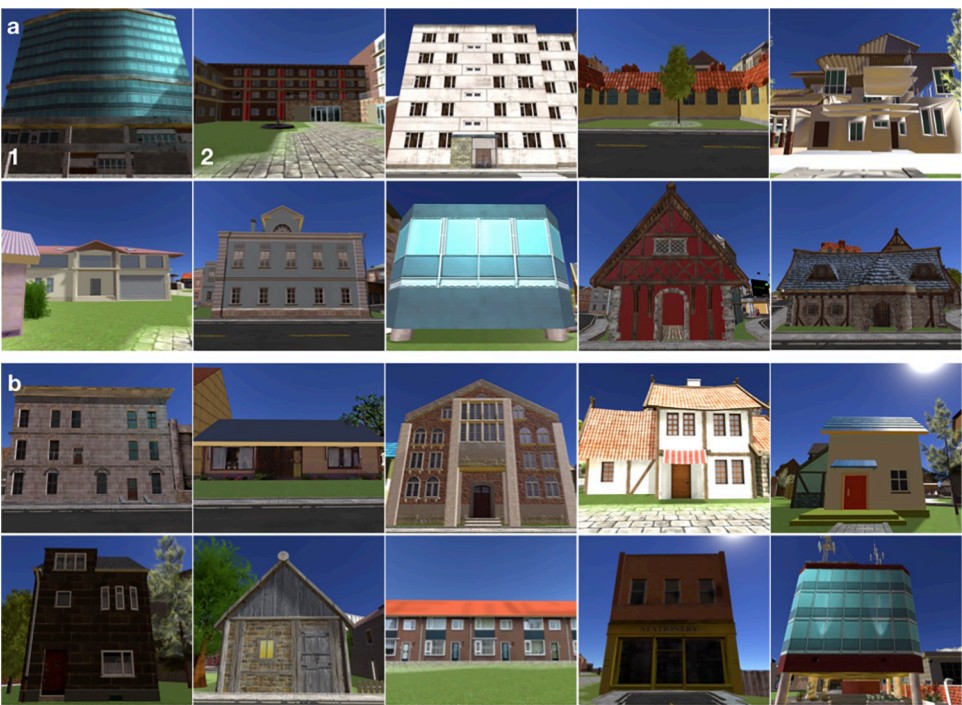

**Fig 10. Example buildings in Seahaven. a)** The upper two rows show the ten houses being labeled as gaze-graph defined landmarks. House 1 being the house with the highest and house 2 with the second highest node degree. **b)** The lower two rows show ten other randomly selected houses out of the remaining 203 houses that are located in Seahaven.

city to make all houses unique and varying in style, color and size (Fig 10). The houses were placed randomly in the city without the intention of a city center and city districts. Our results revealed that the gaze-graph-defined landmarks also had features of global and local landmarks.

While it is undisputed that landmarks are important for spatial navigation, the details about their functionality are not yet completely understood. Our results are in line with two possible uses of fixated landmarks in agreement with previous research [2,4,5]. Participants could have fixated a single landmark to use it as either a beacon for goal directed navigation or a visual cue for direction and orientation information [2,6,7]. Looking at two landmarks would allow triangulation. In contrast to trilateration, triangulation does not involve distance measurements to the two reference points. Thus, only with knowledge of the base length, the distance between the two reference points, explicit distances can be determined. Nevertheless, triangulation is widely used in a multitude of applications, including spatial navigation. Our investigation of a triangulation revealed that in 71.9% of the visited city area at least one gaze-graph-defined landmark was viewed. Furthermore, in 39.1% of the city area that was visited by participants and 50% of their experiment time, two or more gaze-graph-defined landmarks were viewed. Thus, participants could have combined information from viewing two or more landmarks from these locations. Multiple landmarks and knowledge of their interrelations could serve as the basis for an allocentric representation of the environment [2,4,10,11]. This is supported by calculating the rich club coefficient of all gaze graphs. Here, we found a correlation between increasing node degree and increasing connectivity between the respective nodes. Gaze graph-defined landmarks were above chance level visually interconnected to each other. Thus, the interconnected gaze-graph-defined landmarks of the city seem to form a network, i.e., a rich club, covering a large, mostly central part of the city. This could be a further

indication that landmarks are not only used as orientation anchors, but could form an underlying network of orientation anchors that span out the framework of a mental map. Consequently, landmarks could anchor the mental map and thus, serve as an important feature of survey knowledge.

## Outlook & conclusion

While the application of graph theory has enabled us to use a variety of already established graph-theoretical measures to analyze the gaze graphs and resulted in very promising results, only a small amount of the available graph-theoretical measures was applied during our analysis. For example, the node degree centrality is defined as the sum of connections of each node. Generally speaking, nodes with a lot of connections are likely to be important for most networks [53] which is why the node degree centrality is widely used in graph-theoretical analyses and serves as the basis of our gaze graph analysis [60]. According to Sporns [53], the usage of graph theory in neuroscientific studies has increased in the recent years and the node degree centrality can be a useful measure for network analyses. However, these analyses are mostly based on pairwise dyadic approaches and the full potential of graph theory has not yet been applied.

A variety of graph measures is available for analyzing networks and node importance can be defined in different ways, exceeding the node degree-based analysis. A particularly important measure in social network analyses is the so-called betweenness centrality [61]. The measure counts how many shortest paths cross a particular node that is "between" a lot of nodes. In social network analyses, betweenness centrality is beginning to replace the node degree centrality to explain social network dynamics with respect to the importance of nodes with high node betweenness for attracting and strengthening new links [62]. Consequently, the betweenness centrality could be a potential candidate for explaining spatial knowledge acquisition. As mentioned, the centrality measure gains importance within the field of social network analysis and is used to characterize the attraction and strengthening of connections. A person, represented by a node in a social network, would have a high betweenness centrality if the person connects a variety of other persons that themselves do not know the connecting person. By transferring this thought to spatial navigation, a house with high betweenness centrality would connect the views to two buildings that are not viewed in direct succession themselves. Thus, this building serves as an anchor point for these two buildings and forms a part of the route between them. Betweenness centrality could serve as a measure for characterizing the gathered route knowledge. In short, graph theory offers a variety of different measures and has not yet reached its full potential within neuroscientific research.

Overall, our results establish a new methodology to process eye tracking data in complex 3D environments and identify and assess the function of landmarks in spatial navigation. Applying this methodology provides a new and unique insight into behavioral data of visual attention processes during spatial navigation and opens the door for a novel approach to investigate spatial navigation. To fully unlock the potential of graph theory, we propose additional graph-theoretical measures to investigate gaze graphs in the future. Specifically, we consider the betweenness centrality that could help to understand the formation of spatial knowledge beyond landmark knowledge.

## Methods

### Ethics statement

At the beginning of the experiment participants were informed about the investigation's purpose and procedure, and gave written informed consent. Each participant was reimbursed

with nine euros per hour or earned an equivalent amount of "participant hours", which are a requirement in most students' study programs at the University of Osnabrück. Overall, each session took about 2 hours, thus accumulating to about 6 hours over all three sessions. The Ethics Committee of the University of Osnabrück approved the study following the Institutional and National Research Committees' ethical standards.

## The virtual town of Seahaven

The virtual town of Seahaven was built to investigate spatial learning during free exploration of the virtual environment and compare it to learning of a city map [34–36]. Seahaven is a fictional place and was created for the specific purpose of that experiment with the Unity game engine using buildings, streets, and landscape objects acquired from the Unity asset store. The city was designed as a connected urban space of roughly 216,000 m$^2$ [36]. In total, Seahaven contains 213 unique houses with individual appearance of style, shape and size (Fig 10). The houses were randomly placed in the city to avoid typical city districts or clusters of houses with outstanding features as well as avoiding a urban design with a city center and surrounding suburbia areas (for more details see [34–36]). The street structure consisted of winding and straight, small and big roads, and overall resembled a small European city. The entire virtual environment reflects natural spatial relations, with one Unity unit corresponding to one meter. The virtual height of the participant was set to 2 meters and was unified for every participant. To provide a better frame rate in Unity, a far clipping plane in a distance of 160 meters was introduced. Consequently, no objects were visible to the participants that were located further than 160m away from current participant location.

## Structure of the experiment

In total, 22 participants took part in the experiment for the present publication. The experiment consisted of three repeated sessions within 10 days of the first session. Each session consisted of five parts. First, participants received a brief introduction to the experiment and signed the necessary paper work. In the second part, participants were introduced to the type of spatial tasks they would need to complete after the exploration session of the virtual city. The spatial tasks included judging the orientation of a house towards north, judging the orientation between houses, and judging the direction between houses using a pointing task. All tasks were performed in two response conditions (within 3 s and infinite time) and designed as two alternative forced choice (2 AFC) tasks outside the virtual environment. Participants were then asked to complete task examples with test stimuli using photographs of houses of the city of Osnabrück. The third part was the setup of the participants for the exploration of the VR environment. This consisted of the adjustment of the VR-headset including the eye tracker and the calibration and validation of the eye tracker. In the 1$^{st}$ session it was complemented by a movement training on a separate virtual island to get participants accustomed to the possible movements in VR. In the fourth part, the main experiment took place. Here, participants explored the virtual town for 30 minutes while movements and eye tracking data were recorded. Participants were instructed to freely explore the city as if they would explore a new real-world city for the first time. In the fifth and final part of the experiment, participants performed the three spatial tasks (see above) outside of the VR using images of houses of the virtual city Seahaven. A complete description of the experimental procedures can be found in [35,36].

## Laboratory setup

Participants wore a head mounted HTC Vive virtual reality headset and were seated on a swivel chair. To prevent limitations on rotations as well as removing the tactile directional

feedback due to hanging cables, a vertical cable solution was implemented. Participants moved using the HTC controller at walking speed. To decrease the risk of motion sickness, participants were instructed to only walk forward with the controller. If they wanted to switch directions or turn, they were instructed to stop walking and then rotate their entire body with the chair in the desired direction.

## Eye tracking

A pupil labs eye tracker (gaze accuracy 1.0˚, gaze precision 0.08˚) was directly integrated in the HTC Vive headset (screen resolution: 1080 x 1200 pixels per eye (2160 x 1200 pixels combined), screen refresh rate: 90 Hz, field of view: 110 degrees)[34]. Both calibration and validation were executed in the virtual reality while the participants where still on a separate training island. For each participant, a 17-point calibration and a 9-point validation were conducted until the validation error was below 2˚ (on average 1.53˚). During the experiment, a 1-point validation was performed every 5 minutes ensuring the correct tracking precision. If the error exceeded 2˚, a complete 17-point calibration and 9-point validation was performed until the error was below the original threshold of 2˚ (mean 3.14˚, median 2.11˚ before and mean 1.88˚, median 1.28˚ after a new calibration). 55% of all 1-point validations had a validation error above 2˚ and had to be recalibrated, hence, highlighting the importance of regular validation control. When the calibration and validation process was performed during the exploration of Seahaven, the display of the virtual city disappeared until the process was completed.

## Graph Theory – A detailed explanation of used graph measures

In the following, we describe the graph theory measures that are used in this publication on examples of simple networks.

## Node degree centrality

The main measure of this publication is the node degree centrality. In the field of graph theory, the node degree gives the total number of connections (edges) of one node to any other node. An isolated node has a degree of zero. The maximal degree occurs when a node is connected to all other nodes, i.e., it acquires a node degree of the number of nodes in the graph minus one. When calculating the node degree centrality for every node in the network, the node degree distribution can give insights into the general connectivity of the network or to identify outliers i.e., nodes with extremely high or low node degree. For example, the node degree distribution of the graph shown in Fig 11A has a mean node degree value of 3.82+/-1.8. Therefore, a 1-sigma distance to the mean at 5.62 and a 2-sigma distance to the mean at 7.42. We have one node with a degree larger than 7, which thus seems to be especially well connected. It can be examined as the only yellow node (Fig 11B). Overall, node degree centrality is rather fundamental and one of the most widespread used measures.

## Graph partitioning

Partitioning investigates whether a graph can be considered as one connected graph or clearly distinguishable subgraphs. As an example, we applied the spectral partitioning approach to a graph shown in Fig 12. It has two nearly fully connected subgraphs that are interconnected by two edges only (Fig 12B). The spectral partitioning approach includes three steps: the calculation of the graph's Laplacian matrix, finding the second smallest eigenvalue, and splitting the graph based on the corresponding eigenvector. First, we consider the adjacency matrix and the degree matrix of the graph. The former is a binary square matrix where each entry indicates

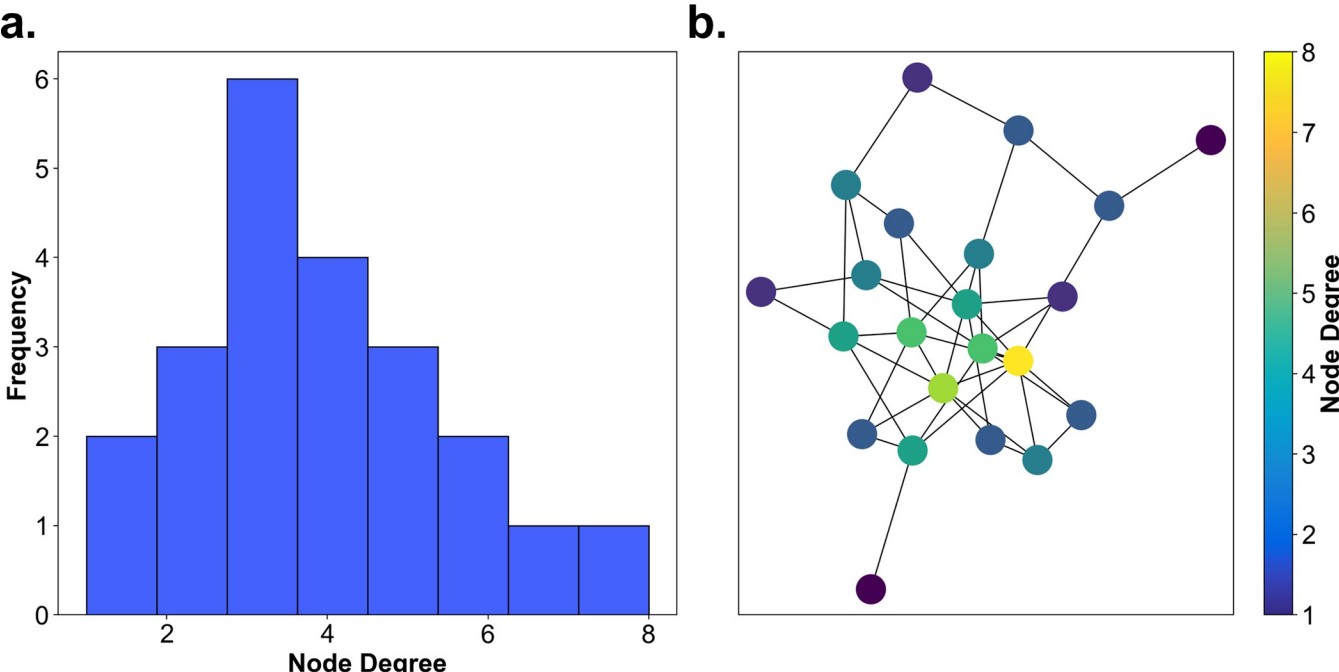

**Fig 11. Node degree–Toy example. (a)** The node degree distribution of an example graph with similar distribution to the graphs used in the publication. **(b)** The example graph with color-coded nodes according to their individual node degree defined by the number of edges that connects a node to other nodes.

whether an edge connects the two nodes / houses. The latter is a square matrix that contains the degree of each node on its diagonal and is zero otherwise. We calculate the Laplacian matrix by subtracting the degree from the adjacency matrix. Second, according to [41,42], the spectrum of a graph, i.e., the eigenvalues of the Laplacian matrix, can be used as a measure of graph connectivity. We calculate these eigenvalues $\mu_i$ and sort them in ascending order. They have the properties that $\mu_1 = 0$ and $\mu_2 \geq 0$. If the graph is connected the second property holds strictly with $\mu_2 > 0$, which is the case in our analysis ($\mu_2 = 0.5858$). Third, we consider the eigenvector corresponding to the second smallest eigenvalue, the so-called Fiedler vector. It is an approximate solution to the optimization problem of finding the minimum cut that is the minimum number of connections that have to be cut to separate two clusters of nodes. When calculating the eigenvector of a graph's eigenvalue, each entry corresponds to a node of the graph. Thus, we map each node onto the corresponding entry of the Fiedler vector. Then we sort the adjacency matrix of the example graph according to the sorting index of the Fiedler vector (Fig 12C). The color coding indicates the distinction into the two subsets with the two interconnecting edges plotted in black (note that there are 4 dots because the adjacency matrix is a symmetric matrix and each edge appears twice. The block structure is clearly visible and indicates clustering. The partitioning of the graph is performed by splitting of the Fiedler vector into positive and negative parts (Fig 11C) and assigning the corresponding nodes to the two separate groups. The larger the gap between positive and negative values, the fewer inter-cluster connections are present.

### Hierarchy index

The hierarchy index characterizes hierarchical configurations within networks. More specific, the hierarchy index describes how fast the frequency of nodes of certain degree drops when

                                     

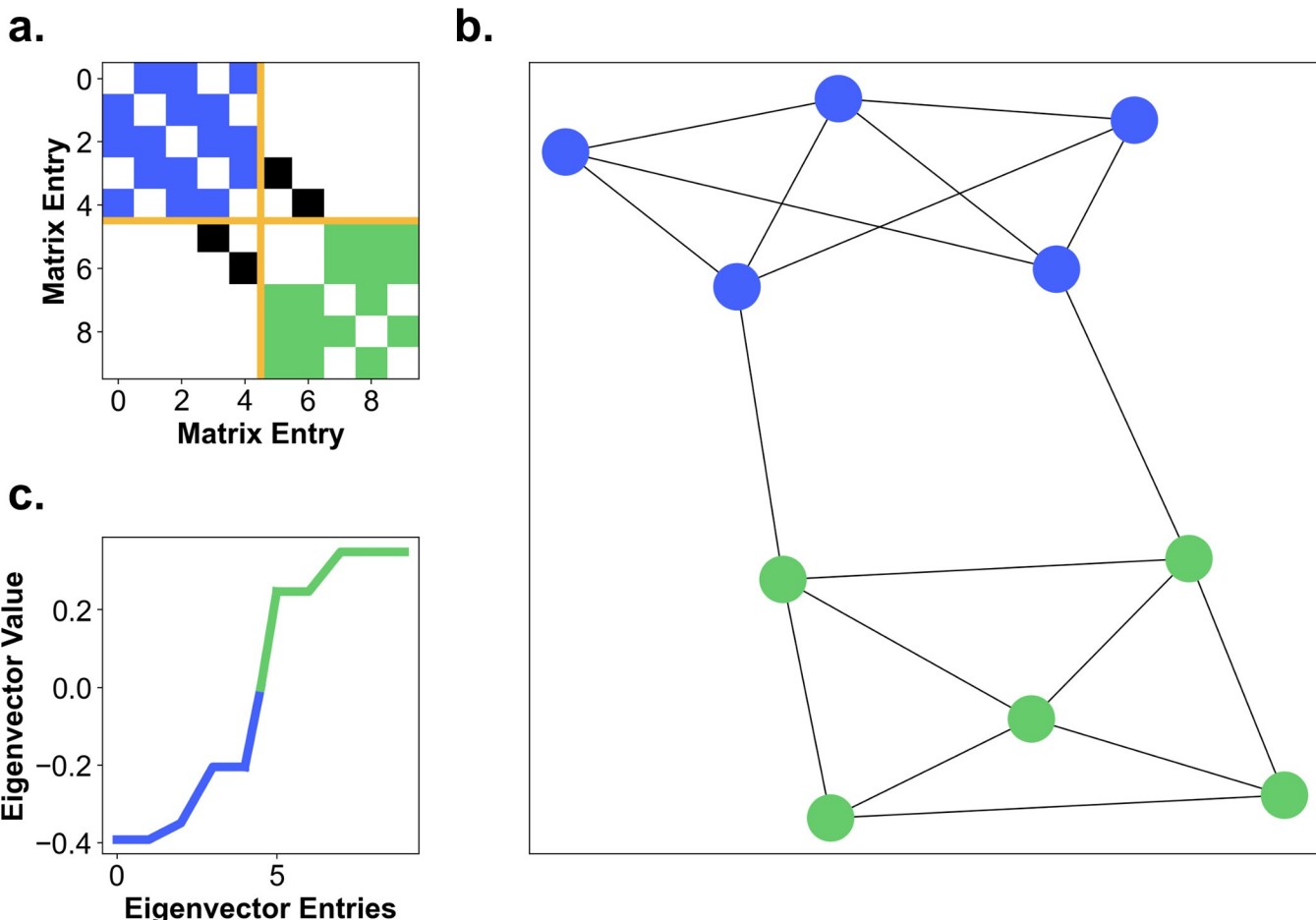

**Fig 12. Graph partitioning–Toy example. (a)** The sparsity pattern of the graph's adjacency matrix sorted by its second smallest eigenvector. Color coded into edges between nodes of one cluster (blue), edges between nodes of the other cluster (green), edges between nodes of the two clusters (black) and a distinction between the clusters (yellow). **(b)** The toy graph with two color coded clusters. **(c)** The second smallest eigenvector of the Laplacian matrix is sorted ascendingly and color coded into two clusters by division into positive and negative components.

increasing the degree. Here, we calculated the hierarchy index for an example graph that was designed to have a hierarchical configuration, but is still similar to the Seahaven graphs. Thus, it has a low number of nodes with a node degree of 1, a larger number of poorly connected nodes with a node degree of 2 or higher, and a small number of highly connected nodes (Fig 13A and 13B). To calculate the hierarchy index, first individual degree values are plotted against their frequency of appearance on a double logarithmic plot (Fig 13C). Then, the data points above the median of the degree distribution are linearly fitted (green line in Fig 13C). In our toy example the median is 4 with a logarithm of 1.39. The hierarchy index is the negative slope of this fitted line and for our toy example results in 2.40. Values above 1 are considered a strong hierarchical configuration.

## Rich club coefficient

The rich club coefficient investigates, whether high node degree nodes are preferentially connected to each other, and, thus, form a rich club. Here, we designed the example graph to have a dense core of 5 highly interconnected nodes and several low degree nodes on the outside

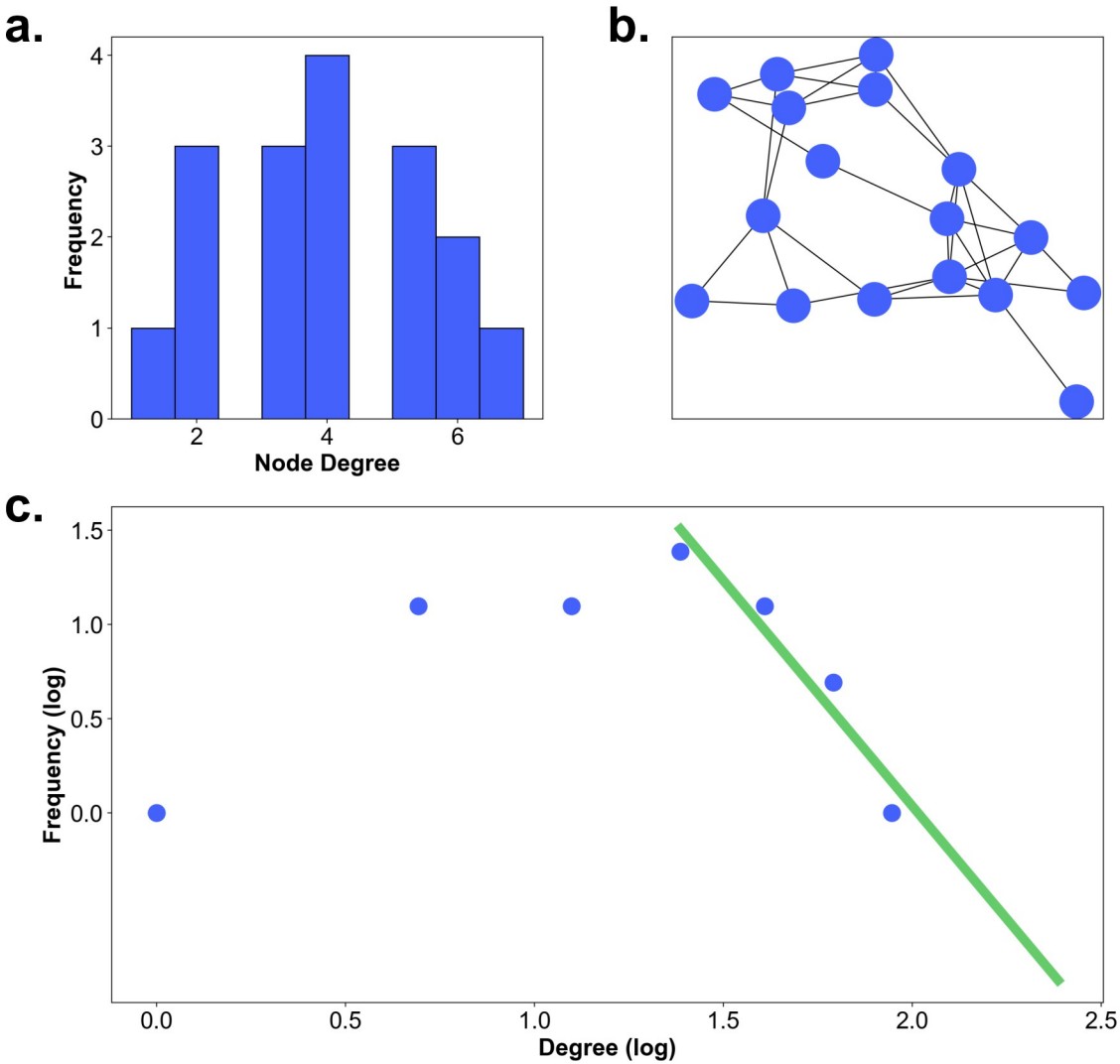

**Fig 13. Hierarchy index–Toy example. (a)** The node degree distribution of the example graph. **(b)** The example graph **(c)** A scatter plot of the logarithm of degree against the logarithm of the frequency of occurrence. The green line shows the linear fit to the part of the data above the median degree.

(Fig 14B). Fig 14A shows the node degree distribution of the graph. It is obvious that the five central high degree nodes have degrees of at least 5, while the rest of the nodes have a degree below 4. The rich club coefficient is based on the relation of edges in the graph in comparison to the number of possible edges that could be instantiated. Therefore, the first step is the calculation of the number of edges of nodes with a node degree of at least 1, i.e., all edges of the graph, and dividing it by the maximum possible number of edges (see the rich club formula in result section The connections between the gaze-graph-defined landmarks–The rich club coefficient). Afterwards, the degree threshold of 1 is increased to 2 and the number of edges of nodes with a node degree of at least 2 are divided by the maximum possible number of edges between those nodes. This procedure is repeated until there are no nodes left with the specified degree. However, to be able to interpret these connectivity values, we need a baseline. Therefore, we repeat the same procedure for a random graph that has the same degree distribution as the example graph. Fig 14C shows the rich club coefficient of the example graph divided by

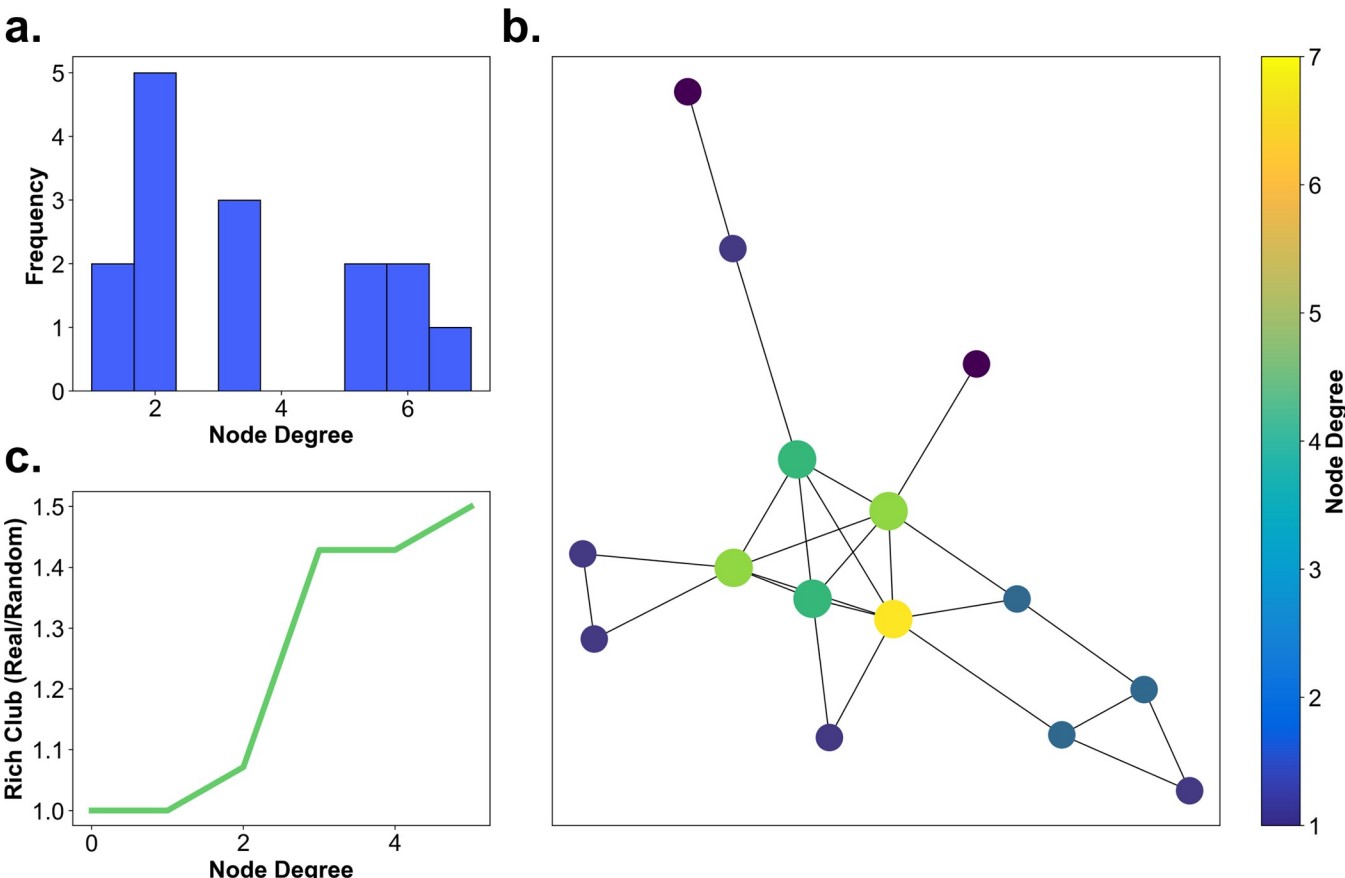

**Fig 14. Rich club coefficient–Toy example. (a)** The node degree distribution of the example graph. **(b)** The example graph with color-coded nodes according to their node degree. **(c)** The adjusted rich-club coefficient, i.e., normalized using a random graph with similar node degree centrality statistics.

the rich club coefficient of a random graph with the same node degree distribution. In the low degrees, the adjusted rich club coefficient has a value of 1, which indicates that this subset of nodes is not preferably connected. However, with increasing degree, low degree nodes are excluded from the rich club calculation. Thus, an increasing slope with increasing degree indicates that the subset of high degree nodes is preferably connected within itself compared to a random graph with the same node degree distribution.

## Acknowledgments

We gratefully thank Viviane Clay and Nicolas Kuske for their assistance and support.

## Author Contributions

**Conceptualization:** Peter König.

**Data curation:** Jasmin L. Walter, Lucas Essmann.

**Formal analysis:** Jasmin L. Walter, Lucas Essmann.

**Funding acquisition:** Peter König.

**Investigation:** Jasmin L. Walter, Lucas Essmann, Sabine U. König, Peter König.

**Methodology:** Jasmin L. Walter, Lucas Essmann, Sabine U. König, Peter König.

**Software:** Jasmin L. Walter, Lucas Essmann.

**Supervision:** Sabine U. König, Peter König.

**Validation:** Jasmin L. Walter, Lucas Essmann, Sabine U. König, Peter König.

**Visualization:** Jasmin L. Walter, Lucas Essmann.

**Writing – original draft:** Jasmin L. Walter, Lucas Essmann, Sabine U. König, Peter König.

**Writing – review & editing:** Jasmin L. Walter, Lucas Essmann, Sabine U. König, Peter König.

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
