## [Decision Letter · Decision Letter 0]

8 Nov 2021

Dear Ms. Walter,

Thank you very much for submitting your manuscript "Finding landmarks – An investigation of viewing behavior during spatial navigation in VR using a graph-theoretical analysis approach" for consideration at PLOS Computational Biology.

As with all papers reviewed by the journal, your manuscript was reviewed by members of the editorial board and by several independent reviewers. In light of the reviews (below this email), we would like to invite the resubmission of a significantly-revised version that takes into account the reviewers' comments.

We cannot make any decision about publication until we have seen the revised manuscript and your response to the reviewers' comments. Your revised manuscript is also likely to be sent to reviewers for further evaluation.

Sincerely,

Wolfgang Einhäuser

Deputy Editor

PLOS Computational Biology

Reviewer's Responses to Questions

**Comments to the Authors:**

Reviewer #1: Review on Walter et al. “Finding landmarks—An investigation of viewing behavior during spatial navigation in VR using a graph-theoretical analysis approach”

Over evaluation

The manuscript by Walter et al. exploits graph-theoretical methods for a quantitative analysis of viewing behavior during a complex spatial navigation in a VR environment. The approach is very innovative and may be looked upon as a pioneering study of quantitative research in navigation. Among other important new findings, results lead to a new data-based account of the concept of visual landmarks.

Comments

1) The introduction is succinctly written and provides most of the background required to evaluate the authors’ findings.

2) Typo “occular” (l. 135) -- it occurs throughout the manuscript and should be corrected.

3) The concept (l. 139) of the collider should be explain to the average reader.

4) The section on defining gazes in VR is very informative and important. However, to evaluate the potential influence of data preprocessing and treatment of missing data on the results, it is often a strategy to run an analysis with different preprocessing parameters to demonstrate reliability of the findings. It is up to the authors to decide if such an analysis is useful.

5) The graph partitioning is an important analysis for the current manuscript. I would like to encourage the authors to introduce the concepts separately using a few toy examples. This can be done in the Methods. In my opinion, the current manuscript can be read as a methodological tutorial. Adding more information here might increase the impact of the manuscript. The same applies to the hierarchy index as well as to the rich club coefficient. Please provide more introductory examples to explain the concept carefully.

6) The reference “Schaeffer (2007)” is not included in the references.

7) Overall, the discussion is well-balanced, but should come with subheadings, which would make it easier for the reader to follow the thoughts of the authors.

8) While I agree that there are many new findings in the current work, the author should at least point to the temporal/sequential aspects of the navigation problem. For example, we look differently at the same scene when viewing it the second time. It might not be useful to include more analyses, however, potential aspects of the dynamics of navigation and how they relate to the current graph-based approach should be explained.

Typos

l. 103: “behavior”

l. 157: “in all following analyses”

l. 183: “We interpolate missing data”

l. 235: delete comma before “a new house”

l. 265: “The latter is a”

l. 269: “sort by increasing size”

l. 358: “To further”

l. 404: “a 2-sigma threshold”

l. 444: “average highest”

l. 465: “reflects”

l. 518: “systems is often”

l. 552: “in 3D”

l. 559: delete comma before “that” (throughout more occurrences)

l. 601: “study are crucial”

Reviewer #2: This manuscript used free exploration of a virtual town to investigate how gaze behaviors can inform spatial navigation. The authors used a novel data analysis technique to categorize gaze behavior in 3D VR and they also applied various graph theory analyses to their eye tracking data. The main findings were that participants viewed the city as a whole entity rather than separate smaller entities, participants on average used 10 houses in the city as landmarks, and these 10 houses were highly interconnected with other houses that were not categorized as landmarks (i.e., formed a rich-club). Participants also spent more than half of their exploration time in view of 2 or more of the 10 landmark houses. The authors use their results to conclude that participants seek out landmarks and strategically choose landmarks that will allow them to navigate around most of their chosen environment. They also discuss the challenges in practicality and data analysis with ecologically valid mobile eye tracking studies. Finally, they suggest using further graph theoretical measures to expand upon their results.

The motivation for the study is compelling because using eye tracking in realistic VR situations is instrumental for discoveries that are ecologically valid. The methods used to categorize gaze were novel and can be very useful for future studies. The choice of using graph theoretical measures seems relevant for the task and seems like a good way to visualize very large quantities of eye tracking data. I think the results are very interesting, and help advance our understanding of how people learn landmarks during navigation, though I think the conclusions the authors draw do not quite reflect the data. I am left with some questions about the methods and results, but with some revisions, I believe this paper would be a valuable addition to the literature.

Major:

1. The main finding of the study is that participants on average chose 10 houses as landmarks and in the discussion, the authors connect that to participants seeking out landmarks and strategically choosing their placement. Another alternative explanation is that more distinctive landmarks were placed around the city among some non-distinctive landmarks. This criticism is potentially addressable by adding more information about the houses in the city. Was the number of buildings with unique properties placed evenly throughout the city? Were there other distinctive features on nearby buildings that participants could have chosen as a landmark? Can this issue please be addressed in some form by the authors in terms of the features of landmarks that are within the rich-club vs. those that are not?

2. The first issue hits on another issue related to the framing of the manuscript. The author bring up an antiquated and largely abandoned hierarchical model of spatial processing, the stage model from Siegel and White (Siegel and White 1975). It is not clear why this model is raised because the authors are not explicitly testing it (nor some of the other models mentioned in the introduction). The focus of the paper is on landmarks, an issue of practically universal consensus in terms of importance within models of spatial navigation, and not routes or survey knowledge. Thus, the intro should probably be reframed to reflect the focus on landmarks and not theoretical models of navigation, which are not directly tested here. In addition, the authors bring up “triangulation.” However, as mentioned in the first point, it is not clear whether subjects are simply looking at salient landmarks or actually looking at multiple (three or more) landmarks, which would be the requirement for something being “allocentric” (Ekstrom and Isham 2017). While the looking patterns are connected amongst landmarks, the analyses in the paper can’t really distinguish looking at three or more landmarks as part of “triangulation” or simply using a single landmark as a beacon or orienting cue. This issue should probably be clarified, unless the authors are able to perform additional analyses that can show conclusively 1) subjects are not simply fixating salient (compared to non-salient) landmarks 2) subjects are consistently fixating three or more landmarks when they are viewing “connected” landmarks within the rich-club, and then using this to orient, in some form.

Minor/Detailed concerns:

3. The authors don’t specify clearly if Seahaven is a real place or if it was a fictional place created for the purposes of this experiment. I suspect that it is not a real place based on the description but having that clarification would be nice.

4. What were the specific instructions given to the participants for the task? There is very little information about the actual exploration. As I understand it, participants were not given any specific tasks during the exploration period except to explore the entire environment. Is that correct?

5. I think the authors go too far with their assumption that participants seek out landmarks, especially in strategic positions. Without knowing more about the configuration of the houses and if the salient features on the houses were equally distributed, I cannot accept this conclusion. Seeking out landmarks also implies that participants are consciously choosing landmarks to orient themselves but I do not think the data can inform us about incidental vs. intentional learning in this task given that participants could freely explore the environment and did not have to memorize the positioning of houses, nor engage in any kind of knowledge test of their environment after.

6. On line 163, the authors define a time scale of “266 ms which is equivalent to eight data samples”. I am unsure how the authors calculated this specific value.

7. In Figure 2b, the authors combine “all cluster durations longer than 1000 ms” in the last bin. I am unsure how this enhances the visualization of the graph. What information does this last bin represent? And how far after 1000 ms are there existing samples?

8. On lines 258-259, the authors mention connections that are maintained and connections that are cut. I am not very familiar with graph partitioning and a description of what each of these terms mean would help my understanding of the analysis.

9. In the graph partitioning section, I do not understand why the second smallest eigenvalue is used for all the further analyses. I also do not know what is gained from splitting the eigenvalues into positive and negative numbers and what the interpretation is for these numbers. An explanation to clarify each of these would be helpful.

10. The font in Figure 5 b-e is very small and hard to read.

11. In Figure 5b, were the participants ordered by node degree? I didn’t see anything in the text about the participants being ordered for this graph but the way the data steadily increase makes me suspect they are in order.

12. I am confused by Figure 8a. The authors describe that “a value above 1 indicates the existence of a rich club with the respective node degree”. The figure shows that values of node degrees have a value above 1, which would imply that all node degrees are a rich club instead of just the 10 houses that are most highly connected.

13. The current data show the top 10 most connected houses averaged over all participants but I would be interested in seeing which houses are most connected for each participant, perhaps to investigate any individual differences.

14. The paragraphs in the discussion about using mobile eye tracking to study more ecologically valid situations does not seem to flow in the discussion. It sounds more like the motivation that fueled the study and perhaps should be integrated into the introduction.

15. The results showed that participants spent more than half of their time in the heart of the city instead of the outskirts, which the authors interpret as meaning that participants would rather be able to see 2 or more landmarks so they could triangulate their position. An alternative interpretation is that participants were simply more interested in staying in the middle of the city because city centers tend to have a lot of interesting things to explore.

Very minor grammar concerns or typos:

1. Line 27: “allows to quantify” -> “allows one to quantify”

2. Line 66: “the own” ->“one’s own”

3. Line 180: “ensures to capture” -> “ensures that most fixations are captures while…”

4. Line 265: “latter a” -> “latter is a”

5. Line 267: “degree-“ had an extra –

6. Line 274: “visible in form” -> “visible in the form”

7. Line 358: “o” -> “To”

8. Lines 364-365: “For the example participant reported already above this results in a slope of -2.6349”. This sentence is confusing to me.

9. Line 419: “allows to quantify” -> “allows one to quantify”

10. Lines 455 and 456: “infer the own” -> “infer one’s own”

11. Line 506: “a virtual reality” -> “a virtual reality task”

12. Line 510: “allows to perceive” -> “allows one to perceive”

13. Line 554: “allows to cleanly separate” -> “allows one to cleanly separate”

14. Line 602: “a few millions” -> “a few million”

References

Ekstrom, A. D. and E. A. Isham (2017). "Human spatial navigation: representations across dimensions and scales." Current Opinion in Behavioral Sciences 17: 84-89.

Siegel, A. W. and S. H. White (1975). The development of spatial representations of large-scale environments. Advances in child development and behavior. H. W. Reese. New York, Academic.

**Have the authors made all data and (if applicable) computational code underlying the findings in their manuscript fully available?**

Reviewer #1: Yes

Reviewer #2: **No: **could not find this

PLOS authors have the option to publish the peer review history of their article (what does this mean?). If published, this will include your full peer review and any attached files.

Reviewer #1: No

Reviewer #2: No
---

## [Decision Letter · Decision Letter 1]

7 Mar 2022

Dear Ms. Walter,

Thank you very much for submitting your manuscript "Finding landmarks – An investigation of viewing behavior during spatial navigation in VR using a graph-theoretical analysis approach" for consideration at PLOS Computational Biology.

There is a final comment by reviewer 2, which should be addressed; once this is done, the manuscript can be accepted without further review. Reviewer #2 apparently had trouble downloading your data (see comment below), I think I could access everything through the links provided in the data availability statement, but it might be good to doublecheck.

Sincerely,

Wolfgang Einhäuser

Deputy Editor

PLOS Computational Biology

[LINK]

Reviewer's Responses to Questions

**Comments to the Authors:**

Reviewer #1: I would like to thank the authors for their revised version. Congrats!

Reviewer #2: #7. In Figure 2b, the authors combine “all cluster durations longer than 1000 ms” in the last bin. I am unsure how this enhances the visualization of the graph. What information does this last bin represent? And how far after 1000 ms are there existing samples?

**Have the authors made all data and (if applicable) computational code underlying the findings in their manuscript fully available?**

Reviewer #1: Yes

Reviewer #2: **No: **I dont think the data are downloadable

PLOS authors have the option to publish the peer review history of their article (what does this mean?). If published, this will include your full peer review and any attached files.

Reviewer #1: No

Reviewer #2: No

Figure Files:

Data Requirements:

Reproducibility:

References:

---

## [Editor Report · Decision Letter 2]

26 Apr 2022

Dear Ms. Walter,

We are pleased to inform you that your manuscript 'Finding landmarks – An investigation of viewing behavior during spatial navigation in VR using a graph-theoretical analysis approach' has been provisionally accepted for publication in PLOS Computational Biology.

Best regards,

Wolfgang Einhäuser

Deputy Editor

PLOS Computational Biology

---

## [Editor Report · Acceptance letter]

29 May 2022

PCOMPBIOL-D-21-01690R2 

Finding landmarks – An investigation of viewing behavior during spatial navigation in VR using a graph-theoretical analysis approach

Dear Dr Walter,

I am pleased to inform you that your manuscript has been formally accepted for publication in PLOS Computational Biology. Your manuscript is now with our production department and you will be notified of the publication date in due course.

With kind regards,

Andrea Szabo
